# Pulmonary Application of Novel Antigen-Loaded Chitosan Nano-Particles Co-Administered with the Mucosal Adjuvant C-Di-AMP Resulted in Enhanced Immune Stimulation and Dose Sparing Capacity

**DOI:** 10.3390/pharmaceutics15041238

**Published:** 2023-04-13

**Authors:** Thomas Ebensen, Andrea Arntz, Kai Schulze, Andrea Hanefeld, Carlos A. Guzmán, Regina Scherließ

**Affiliations:** 1Department of Vaccinology and Applied Microbiology, Helmholtz Centre for Infection Research (HZI), Inhoffenstr. 7, 38124 Braunschweig, Germany; 2Department of Pharmaceutics and Biopharmaceutics, Kiel University, Grasweg 9a, 24118 Kiel, Germany; 3Merck KGaA, Frankfurter Straße 250, 64293 Darmstadt, Germany

**Keywords:** mucosal vaccination, adjuvant, nanocarrier, c-di-AMP, antigen dellivery

## Abstract

The most successful medical intervention for preventing infectious diseases is still vaccination. This effective strategy has resulted in decreased mortality and extended life expectancy. However, there is still a critical need for novel vaccination strategies and vaccines. Antigen cargo delivery by nanoparticle-based carriers could promote superior protection against constantly emerging viruses and subsequent diseases. This should be sustained by the induction of vigorous cellular and humoral immunity, capable of acting both at the systemic and mucosal levels. Induction of antigen-specific responses at the portal of entry of pathogens is considered an important scientific challenge. Chitosan, which is widely regarded as a biodegradable, biocompatible and non-toxic material for functionalized nanocarriers, as well as having adjuvant activity, enables antigen administration via less-invasive mucosal routes such as sublingual or pulmonic application route. In this proof of principle study, we evaluate the efficacy of chitosan nanocarriers loaded with the model antigen Ovalbumin (OVA) co-administrated with the STING agonist bis-(3′,5′)-cyclic dimeric adenosine monophosphate (c-di-AMP) given by pulmonary route. Here, BALB/c mice were immunized with four doses of the formulation that stimulates enhanced antigen-specific IgG titers in sera. In addition, this vaccine formulation also promotes a strong Th1/Th17 response characterized by high secretion of IFN-γ, IL-2 and IL-17, as well as induction of CD8^+^ T cells. Furthermore, the novel formulation exhibited strong dose-sparing capacity, enabling a 90% reduction of the antigen concentration. Altogether, our results suggest that chitosan nanocarriers, in combination with the mucosal adjuvant c-di-AMP, are a promising technology platform for the development of innovative mucosal vaccines against respiratory pathogens (e.g., Influenza or RSV) or for therapeutic vaccines.

## 1. Introduction

Vaccination by parenteral or mucosal administration represents the most successful and effective medical intervention for preventing infectious diseases [1,2,3]. According to WHO records, more than 600 million vaccine formulations are now administered worldwide yearly by injections, resulting in (i) decreased mortality, (ii) extended life expectancy, and (iii) improved quality of life [4]. Almost all bacterial or viral pathogens enter the body through mucosal surfaces. It is well known that the mucosa represents the largest lymphoid organ of the body since these tissues are continuously exposed to the external environment [5]. For example, respiratory pathogens enter the body through the thin mucosal barrier of the respiratory epithelium, which accommodates immunocompetent cells able to fight invading pathogens [6]. Thus, the mucosa is a target for vaccination as well as having physiological and practical advantages, such as (i) the possibility to reduce colonization by induction of local immune responses, (ii) simplified vaccination logistics and costs, (iii) reduced risk of needle stick injuries, (iv) no risk of transmission of blood borne-diseases, and (iv) painless application. However, the stimulation of mucosal immune responses is often challenging. Therefore, one major goal of vaccine design is the induction of a protective and long-lasting immune response against potential pathogens at mucosal surfaces [7].

Currently, most licensed vaccines are still administered by the parenteral route and fail to elicit protective mucosal immunity. Implementation of a mucosal vaccination strategy would enable the elicitation of both systemic and mucosal immune responses, representing a convenient needle-free alternative to parenteral administration [8,9]. Moreover, mucosal vaccination can also induce cytotoxic responses, which might be critical to achieving protection against intracellular pathogens or for the establishment of therapeutic vaccines against both communicable and non-communicable diseases (e.g., cancer, hypertension, atherosclerosis or diabetes) [10,11]. 

Today, efforts focus mainly on the development of subunit vaccines based on well-defined protective antigens rather than the exploitation of attenuated or inactivated pathogens. These formulations exhibit an improved safety profile, but purified antigens are much less immunogenic. This renders necessary the inclusion of adjuvants in their composition. On the other hand, novel antigen delivery systems may play an important role in vaccine development since they could provide at the same time antigen protection, controlled release and intrinsic adjuvant properties [12]. Biomaterials such as polymers and lipids, which are known to have immunomodulatory properties, are of major interest for this application and have the potential to shape the vaccines of the future [13]. 

For mucosal vaccination, the antigen should be taken up in particulate form, as this mediates local processing and induces a local immune response, whereas soluble antigens or very small particles of 50–100 nm are drained to the lymph nodes and elicit only a systemic response [14]. Nanoparticulate antigen carriers produced from biocompatible polymers [15] are favorable formulations and have been shown to be effective in several studies [14,16,17]. Chitosan is a natural biodegradable polymer. The excellent properties, including biocompatibility, non-toxicity and high charge density, are of high interest in biomedical research. In addition, the muco-adhesivity of chitosan creates immense potential for various pharmaceutical applications [18,19]. It is known that chitosan is able to act both as an adjuvant as well as a matrix for delivery carriers when given by the parenteral route [20,21,22,23,24]. However, these particles have also been studied as a promising delivery system for mucosal vaccination, especially via the oral and nasal route [25]. 

Here, we analyzed preliminary adoptive transfer studies optimized chitosan-based nanocarriers of a specific degree of deacetylation, molecular weight and size for the pulmonic application. These antigen carriers are utilized to enable particulate antigen delivery and uptake in the lung, thereby promoting local processing and subsequent initiation of an adaptive immune response [26,27]. Vaccine preparations should be able to enhance the immunogenicity of the antigen and promote immune responses of adequate strength and quality, as well as induce long-lasting memory, which can be achieved by the incorporation of adjuvants. Adjuvants promote immunogenicity by inducing a pro-inflammatory environment that fosters local recruitment and subsequent activation of antigen-presenting cells (APC) [28]. In addition, adjuvants can enhance antigen processing and presentation, induce cytokine expression by APC and bystander cells, and modulate downstream adaptive immune reactions. In addition, adjuvants are able to act as immunopotentiators with immune stimulatory effects during antigen presentation, resulting in the induction of co-stimulatory molecules on APC [29]. An increased understanding of innate and adaptive immune activation will help to develop adjuvants that are tailored for directing and potentiating antigen-specific immune responses [30].

Even though adjuvants have been used for co-formulation with vaccine antigens for more than 70 years, only a handful has been licensed for human use, such as, e.g., emulsions such as MF59 and AS03, Toll-like receptor (TLR) agonists (CpG or monophosphoryl lipid A (MPL) adsorbed on aluminum salts as in AS04) or combination of immunopotentiators (QS-21 and MPL in AS01) [31]. Moreover, only a few adjuvants are available for mucosal administration [32]. Thus, over the last decade, several mucosal adjuvants have been proposed, such as (i) the cyclic di-nucleotides (second messenger molecules in bacteria and archaea) [33], (ii) the cholera toxin subunit [34,35,36], (iii) CpG oligodeoxynucleotides [37,38,39], and (iv) the macrophage activating lipopeptide (MALP-2) [40,41,42]. Among them, cyclic di-nucleotides, like c-di-AMP, are emerging as very promising candidates. They are STING agonists, which promote the expression of type I IFNs and TNF, thereby leading to the activation and maturation of APC, with subsequent stimulation of humoral and cellular responses, which encompass activation of Th1/Th2 cells, induction of multi-functional T cells and stimulation of CTL [43,44,45,46,47,48,49,50,51,52,53,54,55,56,57,58,59].

From a formulation point of view, most vaccines are prepared as liquids for injection for parenteral administration to avoid physicochemical degradation and loss of biological activity due to enzymatic processes or unsuitable pH values on mucosal tissues. In addition, processes during vaccine production or application are susceptible to damaging the antigen by extreme temperatures (e.g., freezing) or physical stress. These circumstances still present challenges in vaccine formulation and hamper their distribution in developing countries [60]. These problems can be overcome by the formulation of vaccines as a dry powder, which often shows increased thermal stability, allowing storage without a cold chain [61]. Thereby the formulation, a dry powder for inhalation, follows a Nano-in-Microparticle approach. On the nanoparticulate level, the dosage form showed high stability and can be administered by a dry powder inhaler (DPI) without reconstitution [62]. Furthermore, dry powder vaccines can be used for direct application via inhalation to the respiratory tract, thus avoiding any preparation steps, such as reconstitution, thereby allowing the patient to self-administer vaccines without risk [63].

Here we describe the DPI formulation of a chitosan nanoparticulate vaccine that has been optimized with respect to chitosan quality [26,27], and we provide experimental data on its successful in vivo proof of concept following pulmonary application to mice. 

## 2. Material & Methods

### 2.1. Nanoparticle Formation and Characterisation

Nanoparticles (NP) were produced by ionic gelation, as described by Wang et al. [64]. For this, 0.1% (*w*/*v*) chitosan (Heppe Medical Chitosan GmbH, Halle (Saale), Germany) and 1 mg/mL Ovalbumin (LPS-free OVA; Sigma, St. Louis, MO, USA) as model antigen was dissolved in 2% (*v*/*v*) acetic acid. 0.1% (*w*/*v*) of the counter ion sodium carboxymethyl cellulose (Tylose C30, Hoechst, Frankfurt, Germany) was dissolved separately in double distilled water of the same volume and added slowly to the chitosan phase while stirring [64]. The size of the spontaneously formed nanoparticles was determined via photon correlation spectroscopy (PCS) (Zetasizer Nano ZS, Malvern Instruments, Malvern, UK). Protein quantification was performed by Micro-BCA assay (Thermo Fisher Scientific, Waltham, MA, USA) upon dissolution of the nanoparticles. Data are presented as the average of three independent measurements.

### 2.2. Transfer to Dry Powder Nano-In-Microparticulate Formulation

2% (*w*/*v*) mannitol (Pearlitol 200 SD, Roquette, Lestrem, France) was dissolved in the nanosuspension, and the preparation was spray-dried (SD) using the Büchi B-290 Mini spray dryer (Büchi, Flawil, Switzerland) with an inlet of 80° and outlet temperature of 35 °C, respectively. To check whether the nanoparticles can be redispersed from the Nano-in-Microparticulate (NiM) formulation, 21 mg dry powder was redispersed in 1 mL 1% acetic acid and measured with PCS. In an additional experimental design, the spray parameters were varied in terms of spray air flow and spray feed solid content to gain an understanding of tuning possibilities in case particle size needs to be optimized for targeting specific areas in the lung.

### 2.3. Dispersion and Aerodynamic Assessment

The dry powder was analyzed by laser diffraction (Helos, Sympatec GmbH, Clausthal-Zellerfeld, Germany) upon dry dispersion at 3 bar. Further, dispersion and aerodynamic performance were tested with the Cyclohaler^®^, a dry powder inhaler device (DPI). For this, 20 mg of powder was weighed individually into HPMC capsules. The capsule was pierced in the device and dispersed into the Next Generation Pharmaceutical Impactor at an airflow of 100 L/min for 2.4 s equalling 4 L, according to Ph.Eur. 2.9.18. The resulting fine particle fraction FPF_emitted_ represents the percentage of particles <5 µm of the emitted dose as measured by NGI (*n* = 3).

### 2.4. Scanning Electron Microscopy (SEM) Images

The images were taken with the Smart SEMTM Supra 55VP scanning electron microscope and the Zeiss DSM 940 (both Carl Zeiss AG, Jena, Germany). For preparation, the samples were attached to a conductive polycarbonate film (Leit-Tabs, Plano GmbH, Wetzlar, Germany) and coated with gold utilizing a sputter coater (SCD 005, Bal-Tec AG, Balzers, Liechtenstein) for 65 s at 50 mA under argon gas. An acceleration voltage of 2–5 kV was applied to the electron beam. The microscopy distance was between 6–8 mm for all images; the magnification varied between 250–5000×. The SE detector, which registers the backscattered secondary electrons, was used for detection.

### 2.5. In Vivo Vaccination Studies

For the proof of concept study, female 6–8 weeks old C57BL/6 mice were bred at the animal facility of the Helmholtz Centre for Infection Research under specific pathogen-free (SPF) conditions with food and water ad libitum. For vaccination, mice (*n* = 5 to 10) were immunized with Ovalbumin (OVA) or with Chitosan-OVA nanoparticles (30 OVA μg/dose) co-administered with or without 10 μg/dose of c-di-AMP (HZI-research grade quality), CTB (Sigma, St. Louis, MO, USA) or CpG (ODN1826, InvivoGen, San Diego, CA, USA) administered in a volume of 50 μL per dose. On days 0, 14, 28 and 48, the vaccine formulations were administered by s.c. or pulmonary route, as shown in Table 1. For pulmonary administration, a PennCentury MicroSprayer^®^ (PennCentury, Wyndmore, PA, USA) was used. To facilitate pulmonary application, mice were shortly (10 min) anesthetized with Ketamin/Rompun. The tip of the device was gently inserted down the trachea of the anesthetized animal, and 50 µL of an air-free plume of liquid aerosol was administered directly into the lungs. 

All animal in vivo experiments were approved by the ethical board and conducted in accordance to the regulations of the local government of Lower Saxony (Germany; No. 509.4250204017.08). After 68 days, the vaccinated animals were anesthetized with isoflurane inhalation. Blood was taken with glass capillaries from the retro-orbital plexus. Then, animals were sacrificed by CO_2_ inhalation, and spleens were isolated and processed. 

### 2.6. Adoptive Transfer

For analyzing the T helper polarization in vivo, sorted naïve CD4^+^ T cells from OTII/Thy1.1 or CD8^+^ T cells from OTI/Thy1.1 double transgenic animals were adoptively transferred to “normal” C57BL/6 mice via intravenous injection as described in [65,66,67]. Since the transferred T cells express Thy1.1 (CD90.1) and all T cells from normal C57BL/6 express Thy1.2 (CD90.2), the two cell populations can be distinguished by flow cytometry. Recipient mice received 2–3 × 10^6^ CFSE-labeled cells by tail vein injection and were immunized the following day with 3 to 30 μg OVA or chitosan-OVA nanoparticles in an amount equal to 3 to 30 µg OVA co-administered with 7.5 μg c-di-AMP, either by subcutaneous (s.c.) injection, intranasal (i.n.) or pulmonary application as shown in Table 2. For i.n. immunization mice were shortly anesthetized, and animals were allowed to inhale the vaccine formulation dropwise (10 µL/nostril). After 3 or 5 days, the mice were sacrificed, and spleens and draining inguinal or cervical lymph nodes (LN) were taken. Readout was performed by flow cytometry (gating strategy shown in Appendix A). 

### 2.7. Enrichment of Naïve Non-Activated T Cells

The magnetic separation kit MagniSort™ (eBioscience, San Diego, CA, USA) was used for the isolation of mouse CD4^+^ naïve T cells from spleens or LN. The enrichment of the naïve CD4^+^ T cells was performed by negative selection using a biotinylated antibody cocktail (CD11b, CD19, CD24, CD4^+^4, CD4^+^5R, CD4^+^9b, Ly-6G, γδ T cell receptor (TCR) and CD4^+^ or CD8^+^) and streptavidin-coated magnetic beads. Undesired cells are bound by antibodies and then depleted by using magnetic beads that are separated in a magnetic field, leaving untouched CD4^+^ or CD8^+^ naïve T cells in suspension. 

### 2.8. Cell Labeling with Carboxyfluorescein Diacetate Succinimidyl Ester (CFDA-SE) 

Cells were labeled using CFDA-SE for quantification of proliferation and discrimination of undivided and divided cells by flow cytometry [68]. CFDA-SE is used because it can easily pass cell membranes, is converted inside the cell by esterase activity to CFSE, and binds covalently to intracellular molecules. While CFDA-SE has no fluorescent properties, CFSE has the same spectral characteristics as fluorescein, can be excited with a blue laser (488 nm), and is measured in the FITC channel. When the CFSE signal of proliferating cells is displayed as a histogram, as shown in Figure 5A–C, a typical pattern will be observed, where every peak from right to left represents one round of division, starting with undivided CFSE^+^ T cells. CFSE-positive and -negative cell populations were determined by FACS analysis of the single cell suspension derived from the spleen and draining lymph nodes.

### 2.9. ELISA

For the measurement of antigen-specific (anti-OVA) IgG, IgG1, IgG2c and IgG2b antibodies, mice were bled on days 0, 14, 28, 48 and 68, and serum was separated by centrifugation at 3000 g. Sera were stored at −20 °C. To perform ELISA assays, plates were coated with 100 μL of OVA (2 μg/mL) and incubated overnight at 4 °C. The general ELISA protocol is described in [50,51,53,59]. The antibody endpoint titers were shown as the reciprocal of the highest sample dilution that yielded an OD ≥ 2 times higher than the mean value of the blank sample.

### 2.10. ElISPOT

Spleens from mice of vaccinated groups were harvested and disaggregated with cell strainers. To destroy the red blood cells, the pellet was suspended in ACK buffer. After washing, splenocytes were re-suspended in complete RPMI and counted. For IFN-γ, IL-2, IL-17, and IL-4 enzyme-linked immunospot (ELISPOT), splenocytes were seeded in culture plates in triplicates (5 × 10^5^ or 2 × 10^5^ cells/well) and incubated in the absence or presence of 2 μg/mL of LPS-free OVA and an immunodominant MHC class I-restricted OVA peptide (SIINFEKL), respectively, at 37 °C with 5% CO_2_. After 24 h (IFN-γ) or 48 h (IL-2), enzyme-linked immunospot ELISPOT kits (BD Pharmingen, San Diego, CA, USA) were used according to the manufacturer’s instructions. Plates were scanned with a CTL ELISPOT reader, and the colored spots were analyzed using the ImmunoSpot image analyzer software v3.2.

### 2.11. Proliferation

Antigen-specific proliferation of cells from immunized mice was determined by measuring the incorporation of radioactive ^3^H-thymidine (Amersham International, Freiburg, Germany) into the DNA after antigen restimulation. A detailed description of the general protocol is described in [50,51,53,59]. For the restimulation of splenocytes, LPS-free OVA was added to final concentrations of 1, 5 and 10 μg/mL. As a positive control, ConA was used at a final concentration of 5 μg/mL, whereas negative controls were cultured in medium alone. For all conditions, quadruplicates were cultured in the incubator for 96 h. The concentration of ^3^H-thymidine was determined by measuring the radioactivity in counts per minute (cpm) using the γ-scintillation counter 1450 MicroBeta (Wallac 1450, Micro-Trilux, Perkin Elmer, Waltham, MA, USA). Results were presented as stimulation index (SI) ± standard error of the mean (sem). The SI represented the counts per minute (cpm) of the antigen-stimulated samples (1, 5 and 10 µg/mL) divided by the cpm of unstimulated samples (0 µg/mL).

### 2.12. Statistical Analysis

Antibody titers and ELISPOT analysis were compared among different groups of mice by one-way ANOVA parametric and non-parametric, followed by Turkey’s and Dunn’s post-tests. Values of * *p* < 0.05, ** *p* < 0.01, *** *p* < 0.001 and **** *p* < 0.0001 were considered statistically significant. All statistical analyses were performed using the GraphPad Prism 8.4.3 software (La Jolla, CA, USA). 

## 3. Results 

### 3.1. Preparation of Chitosan-OVA Nanoparticles and NiM Formulation Development 

The Nano-in-Microparticle (NiM) formulation for dry powder delivery of the model antigen via inhalation has been optimized for its purpose. The antigen ovalbumin (OVA), in this study, needs to be associated with a nanoparticulate carrier to allow specific immune targeting and cellular uptake [14]. This renders nanoparticle size, antigen content and nanoparticle stability upon redispersion important characteristics. In order to obtain a storage-stable form of the nanoparticulate vaccine, it has been embedded in a dry mannitol matrix. The ratio of antigen to matrix in the final spray-dried product determines the dose in mg powder to be administered and thus needs to be taken into account for administration in an in vivo small animal setup. Finally, the spray drying parameters can be used to tune particle characteristics such as size and morphology at a fixed composition of the formulation. With this, it is possible to fine-tune aerodynamic performance and regional deposition if desired. The preparation of OVA nanoparticles by ionic gelation reproducibly resulted in 209 nm +/− 16 nm nanoparticles (Figure 1A) with an OVA load of 10% (*w*/*w*). Embedding those nanoparticles in a mannitol matrix (5% mannitol) resulted in a dispersible powder with a good yield (~70%) in the spray drying process and a microparticle D_50_ of 3.23 µm (Figure 1B). 

However, the unfavorable ratio of antigen to matrix of 1:500 made it unsuitable for administration in an in vivo small animal setup due to the high powder dose needed for the administration of the effective antigen dose. Therefore, the formulation development aimed at minimizing the matrix component while maintaining good stabilization and powder characteristics, as well as a good redispersibility of the incorporated nanoparticles. Transfer to a dry powder NiM formulation with 2% mannitol led to spherical particles (Figure 2A) of low density and a D_50_ of 1.7 µm ± 0.02 µm (Figure 1B). 

The dry powder showed good aerosol properties resulting in an FPF_emitted_ of 75.8% ± 4.5% and a corresponding MMAD of 1.77 µm ± 0.87 µm. Upon redispersion, the powder released nanoparticles of the same size compared to the nanosuspension before spray drying (Figure 1A), making this formulation suitable for pulmonary application and mucosal vaccination. This formulation is produced from the nanoparticle dispersion by adding 2% mannitol (resulting in a spray feed solid content of 21 mg/mL) at 80 °C inlet temperature and a spray air flow of 470 L/h resulting in an outlet temperature of about 35 °C has been used for the in vivo studies.

Further experiments aimed at the definition of design space for the production of Nano-in-Microparticles (NiM) to allow tailored NiM dry powders of defined particle size, which would be deposited in distinct regions of the lung to target immunocompetent cells in the bronchioles or upper bifurcations for example. Unlike systemic application, lung delivery of a vaccine targets the locations where immunocompetent tissue is present such as the aforementioned places [69,70]. Particles of size between 1.3 µm and 4.5 µm (D_50_) could be produced by adjusting the solid content of the spray feed and utilizing a spray air flow of 470 L/h. This translates to differences in mass median aerodynamic diameter (MMAD) and thus deposition profile while maintaining a high fine particle fraction (FPF) of above 65% (Figure 2B and Figure 3).

### 3.2. The Immunization with Chitosan-OVA Nanoparticle Co-Administered with c-di-AMP Is Well Tolerated

For vaccination, groups of 5 to 10 mice (C57BL/6) were immunized with OVA (30 μg/dose) or with chitosan-OVA nanoparticles (30 μg dose) co-administered with or without 10 μg adjuvant/dose administered in a total volume of 50 μL per dose by pulmonary route on days 0, 14, 28 and 48 (Table 2). We did not observe any signs of acute toxicity in animals receiving chitosan-OVA nanoparticles by pulmonary route, as shown in Figure 4A,B. The use of different adjuvants, such as CTB or CpG, resulted in a decrease in weight within 3 days after each immunization. The decrease seems to be even stronger combining chitosan nanoparticles with either adjuvant. However, all the animals recovered within 5 days after each immunization, and weight loss never reached more than 6% (c-di-AMP) of the starting weight, in opposite to standard adjuvants, such as CTB (>12%) or CpG (>8%), which showed enhanced weight loss after immunization. Interestingly, mice immunized with empty chitosan particles + c-di-AMP also showed only minor weight loss underlining the fact that only the combination of antigen nanoparticle and adjuvant resulted in increased reactogenicity. Nevertheless, the downstream processes of this reactogenicity seem to be based on the co-administered adjuvant, as c-di-AMP showed better tolerability, with no or weaker loss of body weight after vaccination, which was comparable to what was observed in animals receiving OVA co-administered with c-di-AMP or chitosan-OVA nanoparticles by the s.c. route. 

### 3.3. Mucosal Vaccination Strategies Using Chitosan-OVA Nanoparticles 

In vivo effectiveness was assessed in two different setups. The first assessment was performed in naïve mice to look at the immune effect in a proof of principle study. An adoptive transfer model was then used to assess the extent of immune response on an organ-specific level. The adoptive transfer model is a well-established tool for the characterization of T cell activation in vivo since TCR-transgenic mice are tolerogenic to their TCR-specific Ag [70]. For studying novel mucosal vaccination strategies, adoptive transfer experiments with Ag-dependent Th polarization of OTI (CD8^+^) or OTII (CD4^+^) T cells and proof of principle animal studies using chitosan-OVA nanoparticle co-administered with c-di-AMP were performed. For the identification of the transferred TCR transgenic T cells in the wild-type C57BL/6 host, Thy1.1 x OTI or OTII mice were used as donors. The Thy1.1 (CD90.1) congenic marker was used to distinguish between wild-type C57BL/6 and Thy1.1/OTI or OTII cells. The Thy1.2 marker expressed by C57BL/6 is slightly different so non-cross-reactive monoclonal Abs were able to detect each form of Thy1. Transgenic Thy1.1/OTI or OTII T cells were injected into the tail vein of each C57BL/6 recipient mouse. Mice were immunized 24 h later with OVA solution (30 µg) or chitosan-OVA nanoparticles (30 µg) co-administered with c-di-AMP s.c., i.n. or pulmonary route (Table 2, Figure 5E). Stimulation of adoptively transferred OTII CD4^+^ T cells by OVA co-administered with c-di-AMP showed strong responses in different local and systemic tissues when vaccinated by i.n. or pulmonary route (Figure 5A). Chitosan-OVA nanoparticles co-administered with c-di-AMP given by i.n. or pulmonary route showed the strongest cell proliferation in local draining LNs (Figure 5B). 

The application of 30 µg OVA co-administered with 10 µg c-di-AMP induced strong proliferation of Thy1.1^+^/CD4^+^ T cells. Overall, we observed local cell activation in draining LNs and at the site of administration (lung), as well as systemically (spleen) following i.n. or pulmonary administration (Figure 5A,B). In contrast, no proliferation was observed in control animals (Figure 5A). Co-administration of 3 or 30 µg chitosan-OVA nanoparticles with 10 µg c-di-AMP by i.n. and pulmonary route, respectively, resulted in the induction of strong proliferation of Thy1.1^+^/CD4^+^ T cells in the cervical draining LNs shown by the loss of CFSE (Figure 5A,B). On the opposite, samples derived from mice treated with empty chitosan-nanoparticles showed only weak unspecific proliferation (Figure 5B). Interestingly, analysis of lung and spleen samples derived from mice immunized with the nanoparticulated formulation showed only weak induction of cell proliferation with respect to those proliferation rates observed in mice receiving OVA protein + c-di-AMP (Figure 5A). However, vaccination of C57BL/6 mice with chitosan-OVA nanoparticles co-administered with c-di-AMP by pulmonary route resulted in the activation and strong proliferation of adoptively transferred Thy1.1^+^/CD8^+^ T cells at both local (lung and cLNs) and systemic level (spleen). 

Interestingly, the combination of chitosan-nanoparticles with a reduced amount of OVA (here: 3 µg) resulted in similar cell proliferation, indicating the dose-sparing potential, and resulted in vigorous activation of OVA-specific CD8^+^ T cells at the site of administration (lung), as well as systemically (spleen) following i.n. or pulmonary administration as shown in Figure 5C,D. This is of importance for all therapeutic strategies where cytotoxic immune responses are needed (e.g., cancer vaccines). A similar dose-sparing capacity at a lower extent was observed when OVA was co-administered with c-di-AMP (Appendix A). Interestingly, the tight interaction of chitosan and antigen seems to be the base for the strong proliferation shown, whereas a lower immune response was seen when antigen-free NP was co-administered with 30 µg OVA (Figure 5D). Thus, while immunization of mice with OVA-loaded chitosan-nanoparticles resulted in the proliferation of almost all OVA-specific CD8+ T cells as indicated by the high number of CFSE low CD8+ T cells, administration of nanoparticles co-administered with OVA and adjuvant was less efficient (Figure 5D). This shows that the incorporation of OVA during the NP formation process is key to the observed effect.

### 3.4. Pulmonary Administration of Chitosan-OVA Nanoparticles Co-Administered with c-di-AMP Induces Strong Cellular Immune Responses 

The analysis of the lymphoproliferative responses of splenocytes showed that chitosan-OVA nanoparticles co-administered with c-di-AMP (▽) by pulmonary route were able to induce a strong lymphoproliferation after re-stimulation with 1 µg of LPS-free OVA with a statistical significant (*p* < 0.001, ***) stimulation index (SI) of >6, (Figure 6A). In contrast, no OVA-specific proliferation (SI < 3), even with higher dosages (5 to 10 µg), was observed in mice receiving chitosan-OVA nanoparticles alone or co-administered with CTB (△) or CpG (◇). The same is true for mice immunized with empty chitosan-nanoparticles co-administered with CTB by s.c. route (SI < 3). Moreover, no statistically significant proliferation has been observed in these mice (Figure 6A).

The analysis of the lymphoproliferative responses of splenocytes showed that OVA protein co-administered with c-di-AMP (▽) by pulmonary route was also able to induce a strong lymphoproliferation after re-stimulation with 1 µg of LPS-free OVA with a statistically significant (*p* < 0.001, ***) SI > 3 (Figure 6B), whereas OVA co-administered with or without 10 μg/dose CTB (△) showed no statistically significant proliferation. 

In order to receive an initial impression of the type of cellular immune response promoted in the vaccinated animals, cytokines secreted in the supernatants of OVA-restimulated splenocytes were analyzed. Cells from mice vaccinated with a chitosan-OVA nanoparticle formulation co-administered with either c-di-AMP or CpG by pulmonary route secreted predominantly IL-17 and IFNγ when re-stimulated with 1 µg OVA (Figure 7A). This suggests a response polarized towards a Th1/Th17-dominated profile. In addition, pro-inflammatory cytokines, such as IL-6 and TNFα, were mainly secreted in animals receiving chitosan-OVA nanoparticles co-administered with c-di-AMP, while in mice immunized with OVA, no statistically significant differences in IL-6 and TNFα concentrations could be observed with respect to the untreated control group. This is in line with previous findings by Ma et al., demonstrating that chitosan oligosaccharides inhibit IL-6 and TNF-α production by blocking the mitogen-activated protein kinase (MAPK) and the PI3K/Akt signaling pathways [71]. Splenocytes derived from mice immunized with OVA alone or co-administered with CTB by pulmonary route showed only a weak cytokine production. Interestingly, in mice receiving OVA co-administered with c-di-AMP, a higher amount of OVA (10 µg) was needed for the stimulation of comparable levels of IL-17 and IFNγ (Figure 7B), indicating a stronger stimulation of the antigen-specific immune response when OVA was loaded to chitosan nanoparticles. Similar observations have been done before using antigen-loaded nanoparticles for i.n. immunization of mice [72,73]. In mice receiving the chitosan-OVA nanoparticles by the s.c. route, secreted cytokines indicated mixed Th1/Th2 immune responses, as revealed by the enhanced secretion of IFNγ and IL-5.

To further analyze the cellular immune responses stimulated by OVA or chitosan-OVA nanoparticles co-administered with c-di-AMP by pulmonary or s.c. route, ELISPOT assays for antigen-specific IFNγ, IL-17, IL-2 and IL-4 producing cells and lymphoproliferative assays were performed three weeks after the last booster immunization on day 68. Interestingly, while increased cytokine secretion after 96 h could be detected only for splenocytes of mice immunized with chitosan-OVA nanoparticles co-administrated with c-di-AMP, no statistically significant differences could be detected between the numbers of OVA-specific cytokine-secreting splenocytes of mice immunized with chitosan-OVA nanoparticles co-administrated with either adjuvant (Figure 8). Nevertheless, only when c-di-AMP was included as an adjuvant significantly more cells has been stimulated with respect to those stimulated without an adjuvant. Even more remarkable, immunization using OVA protein co-administered with CTB (*p* < 0.0001 (****)) or c-di-AMP (*p* < 0.0001 (****)) was more efficient in stimulating OVA-specific cytokine-secreting cells than the nanoparticulated formulations (Figure 8A–C). Thus, re-stimulation of splenocytes with the OVA CD8^+^ peptide and OVA protein, respectively, for 24 h resulted in the induction of statistically significant (*p* < 0.0001 (****)) larger numbers of IFNγ and IL-2 secreting cells as compared to those observed for the nanoparticulated groups (Figure 8A–C). 

However, it has been demonstrated before that the number of cytokines secreted does not necessarily have to correlate with the number of cells producing these cytokines [74].

The pulmonary application of OVA co-administered with CTB or CpG induced higher levels of IFNγ and IL-2 secretion with respect to values obtained in mice receiving OVA or chitosan-OVA nanoparticles alone (*p* < 0.0001 (****)). Mice receiving the chitosan-OVA nanoparticles by the parenteral (s.c.) route secreted no IFNγ when stimulated with an OVA-CD8^+^-specific peptide, whereas lower levels of IFNγ and IL-2 were observed in response to the full OVA protein.

### 3.5. Chitosan-OVA Nanoparticles Co-Administered with c-di-AMP Elicit Strong Antigen-Specific Humoral Immune Responses 

Antigen-specific IgG1 and IgG2b subclass titers were determined in sera samples derived from vaccinated animals included in a proof of principle animal study at day 68 (Figure 9A,B). There were statistically significant differences (CTB, *p* < 0.05 (*)/c-di-AMP *p* < 0.005 (**)) in the levels of IgG1 and IgG2b titers elicited in mice vaccinated by pulmonary route. Co-administration of CTB or CpG as positive control mucosal adjuvants resulted in the induction of higher levels of IgG2b titers (>6- and >50-fold increase, respectively) with respect to values obtained in mice receiving OVA or chitosan-OVA nanoparticles alone at *p* < 0.05 (*). On the other hand, immunization with OVA co-administered with c-di-AMP resulted in significantly at *p* < 0.005 (**) enhanced OVA-specific IgG1 and IgG2b subclass titers when given by pulmonary route (Figure 9A,B). The analysis of the IgG subclasses induced by vaccination with the chitosan-OVA nanoparticle formulation co-administered with c-di-AMP by pulmonary route also showed an enhancement of both IgG1 (>13-fold increase) and IgG2b subclass titers (>59-fold increase) compared to the non-adjuvanted group receiving the chitosan-OVA nanoparticles alone. 

Additionally, in comparison to animals vaccinated by s.c. route, the analysis of the IgG subclass titers obtained using c-di-AMP as adjuvant showed an increase but less prominent with 2-fold and >3-fold increases for IgG1 and IgG2b, respectively (Figure 9A,B). The fact that chitosan-OVA nanoparticles alone stimulated stronger antigen-specific humoral responses when given by s.c. route rather than by pulmonary route compared to OVA protein alone, most probably because some part of the vaccine formulation in the lung could not pass the mucosa and/or degraded by physicochemical (e.g., pH) or enzymatic processes. In contrast, injection usually results in the application of 100% of the formulation. This highlights once more the necessity of adjuvants being included in mucosal subunit vaccine formulations.

To investigate whether the chitosan-OVA nanoparticle formulation co-administered with c-di-AMP has the capacity to induce local immune responses in various mucosal tissues when given by pulmonary route, OVA-specific IgA titers were measured in mucosal lavages (nasal (NL), bronchoalveolar (BAL) and vaginal lavages (VL)) by ELISA. Weak OVA-specific IgA responses were observed in NL and VL samples of mice receiving OVA or chitosan-OVA nanoparticles co-administered with c-di-AMP. 

## 4. Discussion

### 4.1. Mucosal Administration of Chitosan Nanoparticles 

Vaccination via the mucosal administration route is a promising strategy to elicit efficient immune response both systemically and locally, whereas parental administration of vaccines induces only poor mucosal immunity in general [75]. The current human vaccine market accelerates in the development of alternative administration routes, namely i.n., sublingual (s.l.), oral, topical and pulmonary, whereas the vaginal or rectal route of administration is an alternative for certain applications or pharmacological preparations [76,77]. Besides the i.n. route, pulmonary and s.l. application seems to be very promising (e.g., in terms of acceptance and lack of side effects) for mucosal vaccine administration [63]. However, only a few mucosal vaccines have been approved for the human vaccine market. Reasons for this slow development of effective vaccines was in part related to (i) safety concerns, (ii) poor stability, absorption and immunogenicity of antigens, (iii) the limited number of available technologies, such as mucosal adjuvants and delivery system and (iv) the use of live attenuated vaccines, which were not suitable for certain groups of individuals [78]. 

Chitosan easily forms nanoparticles that can encapsulate large amounts of antigens. This polymer is a safe and natural material able to target the mucosal membrane as a mucosal adjuvant [79]. It was already shown that the i.n. administration of recombinant influenza hemagglutinin (rHA) antigen or inactivated virus with nanoparticles composed of poly-γ-glutamic acid (γ-PGA) and chitosan could induce protective immunity in the respiratory tract [80]. Nevertheless, several issues for effective mucosal vaccination need to be clarified, including the antigen-retention period that enables interaction with the lymphatic system and the choice of the right mucosal adjuvant that is well tolerated, non-toxic and able to induce the required immune response [81]. 

### 4.2. Production and Preparation of Chitosan Nanoparticles and Transfer to a Dry Powder NiM 

Chitosan nanoparticles could be prepared reproducibly by ionic gelation of an acidic solution of low molecular weight chitosan (DDA 90%) with carmellose as counter ion inducing nanoparticle self-assembly. The model antigen OVA was present during particle assembly, leading to protein incorporation (10% antigen load calculated on polymer mass). Nanoparticle size was in the range of 200 nm to 300 nm (Figure 1A), which had proven to be beneficial for uptake in immune cells [76].

In order to achieve good storage stability and a dispersible dry powder, the nanoparticles were embedded in a microparticulate matrix of the sugar alcohol mannitol by spray drying to form Nano-in-Microparticles (NiM). Mannitol was chosen due to its characteristics as non-reducible sugar, not affecting the incorporated protein antigen and resulting in a crystalline matrix of high aqueous solubility. It is known that the distribution of immune cells differs between different lung regions in humans [69], and it is obvious that the smaller the MMAD, the more peripheral particles will deposit in the lung [82]. This would enable the vaccine formulation to reach different areas in the lung and to interact with different immune cells, depending on the powder particle size. 

In spray drying theory, different parameters are known to influence the size of the resulting dry particle, which are related to spray fluid composition or process parameters [83,84]. Atomizing energy is a suitable parameter to influence the resulting particle size distribution. By decreasing atomizing energy from 710 L/h to 300 L/h, the mean size of the powder increases from 1.34 ± 0.01 µm to 4.48 ± 0.05 µm, respectively (Figure 2B). Variation of solid content in the spray feed nanosuspension is also an appropriate tool to control particle size. At constant spray process parameters, the higher the concentration of the spray feed, the coarser the resulting powder (Figure 3A–C). However, decreasing the concentration vastly led to particles of uneven size and morphology and a decline in dispersibility, as visualized by SEM and indicated by an increase in x50 (Figure 3D). As droplet size strongly affects the size of the dried particle, different nozzle diameters also have an influence on particle size. By choosing a larger diameter, the mean size increased, but this effect was seen not to be as pronounced as the previously discussed parameters [85]. Drying temperature also has an effect on the resulting particle size. This parameter was nonetheless kept unchanged as it was known to negatively affect the redispersibility of the nanoparticles and potentially the stability of the incorporated antigen [85].

Altogether, our study showed that the production of Nano-in-Microparticle dry powder products with different D50 by changing spray drying parameters was possible, while the composition of the formulation remained unchanged. For pulmonary vaccination, it can be discussed whether a low MMAD envisaging deep lung targeting is appropriate or whether a uniform distribution reaching alveolar dendritic cells as well as airway mucosa dendritic cells in the bronchioles or even further up towards the Waldeyer ring might be preferred. For the latter, the common definition of FPF would not be useful as this only accounts for the lung dose typically being defined as d_ae_ < 5 µm. Detailed in vivo studies looking at differences in immune response following different regions of vaccine deposition will be needed to provide an adequate answer. Results may then be used for the definition of an appropriate aerodynamic particle size fraction for vaccination via the lung. 

### 4.3. Selection of an Appropriate Mucosal Adjuvant

Mucosal adjuvants, such as c-di-AMP, and alternative routes of vaccination are promising aspects to reduce antigen dose, thereby allowing dose sparing, facilitating implementation in mass vaccination campaigns and resulting in cost savings [86,87]. In addition, these novel vaccination strategies could also help to promote efficient mucosal immunity, potentially mimicking the protection against different viruses. Moreover, this strategy confers not only protection against disease (i.e., symptoms) but also against infection (i.e., colonization) and subsequent horizontal transfer. Svindland et al. evaluated the humoral and cellular immune responses of chitosan and the mucosal adjuvant c-di-GMP co-administered with the influenza H5N1 vaccine given by intranasal route in a murine model [88,89]. Due to the potential side effects observed with other adjuvant classes after i.n. administration (e.g., Bell’s palsy), we tested pulmonary delivery as an alternative application route [90]. 

c-di-AMP has been chosen as we have demonstrated previously that incorporation of this adjuvant in different mucosal vaccine formulations efficiently increases antigen-specific immune responses at the mucosal and systemic levels. Thus, when c-di-AMP was co-administered with nanoparticulated antigens, such as OVA or influenza rHA, for example, via the i.n. route, protective immune responses were stimulated [53,91,92]. The same is true when recombinant proteins have been combined with c-di-AMP [44,45,49]. Finally, we searched out that c-di-AMP constitutes a very robust molecule, which can be lyophilized and incorporated into a dry powder formulation without losing functionality.

The chitosan-OVA nanoparticle vaccine was administered as an aerosol by pulmonary route using the Penn Century Micro Sprayer^®^ device. Taken together, animals receiving chitosan-OVA nanoparticles adjuvanted with c-di-AMP did not show signs of acute toxicity (e.g., runting syndrome symptoms, such as loss of body weight, diarrhoea and ruffled hair), whereas a slight decrease in weight in response to the pulmonary vaccination was observed after co-administration with other mucosal adjuvants, like CTB or CpG (Figure 4A). 

The induction of an antigen-specific humoral immune response was determined in sera samples derived from vaccinated animals. The ELISA results of the IgG1 and IgG2b subclass titers underscored the activity of c-di-AMP as a mucosal adjuvant, given the remarkable high serum IgG subclass titers elicited after pulmonary application in comparison to CTB and CpG (Figure 9A,B). The chitosan-OVA nanoparticle formulation co-administered with c-di-AMP led to an increase in OVA-specific IgG1 (>13-fold increase) and IgG2b titers (>59-fold increase) with respect to that observed in animals receiving the non-adjuvanted formulation chitosan-OVA nanoparticles. Similar results were obtained using soluble OVA as an antigen. Nevertheless, stronger responses were obtained using chitosan-OVA alone administered by parenteral (s.c.) route than by using OVA, thereby demonstrating the superiority of the nanoparticle-based formulations. Besides the strong systemic humoral responses, we also observed slightly enhanced antigen-specific IgA secretion at local and distant mucosal territories. 

We showed that the experimental model antigen OVA formulated together with chitosan nanoparticles co-administered with different mucosal adjuvants, such as CTB, CpG or c-di-AMP given by pulmonary route resulted in enhanced Th1/Th17-dominated immune response (Figure 7A and Figure 8A–C). In mice vaccinated with OVA co-administered with c-di-AMP, we observed a comparable but weaker Th1/Th17-dominated immune response, whereas the use of CTB induced only low concentrations of the discussed cytokines (Figure 7B).

ELISPOT studies in which splenocytes were stimulated with OVA or the OVA-specific Kb (MHC class I)-restricted OVA _257–264_ epitope (SIINFEKL) were used to differentiate between CD4^+^ or CD8^+^ T cell-mediated immune responses. From the analysis of the results (Figure 8A), it can be concluded that OVA and chitosan-OVA nanoparticle formulations co-administered with c-di-AMP have a direct effect on the stimulation of CD8^+^ based immunity, as shown by the induction of IFNγ-secreting cells. Similar results were obtained from mice vaccinated with OVA or chitosan-OVA nanoparticles co-administered with CpG, whereas OVA and chitosan-OVA nanoparticles alone or co-administered with CTB resulted in none or only weak IFNγ secretion when cells were restimulated with SIINFEKL. It is known that c-di-AMP induces strong cellular proliferation in different vaccination models [44,45,93]. In this study, only chitosan-OVA nanoparticles co-administered with c-di-AMP by pulmonary route promoted a significant enhancement of antigen-specific proliferative responses (Figure 6A).

### 4.4. Dose Sparing Capacity of Chitosan-OVA Nanoparticles Co-Administered with c-di-AMP 

One of the most interesting features of chitosan-OVA nanoparticles co-administered with c-di-AMP shown in this work is the dose-sparing capacity when given by the pulmonary route (Figure 5C,D and Appendix A). To this end, we observed comparable cellular proliferation of CD8^+^ T cells in cervical LNs and spleen, as well as in mucosal tissues (lung) in mice vaccinated with chitosan-OVA nanoparticles (3 and 30 µg) co-administered with c-di-AMP (Figure 5C). Analysis of cellular immune responses (e.g., proliferation) of single animals exhibited that 4 of 5 animals vaccinated with chitosan-OVA nanoparticles co-administered with c-di-AMP proliferate strongly, as shown by the complete loss of CFSE-staining, whereas only 3 of 5 mice vaccinated with OVA co-administered with c-di-AMP showed this proliferation capacity (Appendix A). Comparative analysis of the cellular proliferation of mice vaccinated with different concentrations of OVA (3 and 30 µg) co-administered with c-di-AMP showed equivalent immune responses (Appendix A).

### 4.5. Induction of OVA-Specific CD8^+^ T Cell Responses by Chitosan-OVA Nanoparticles Co-Administered with c-di-AMP 

Another interesting feature of our most promising formulation encompassing chitosan-OVA nanoparticles and c-di-AMP is the stimulation of antigen-specific CD8^+^ T cells, as presented here in two independent assays. Using the adoptive transfer model based on OVA-dependent Th polarization of CD8^+^ T cells (OT I), we observed strong cellular proliferation of antigen-specific CD8^+^ T cells derived from draining cervical LN and mucosal lung tissues of OT I mice vaccinated with chitosan-OVA nanoparticles co-administered with c-di-AMP (Figure 5C,D). This is in line with our findings showing that vaccination of C57BL/6 mice with chitosan-OVA nanoparticles co-administered with c-di-AMP also stimulated antigen-specific CD8^+^ T cells. Thus, re-stimulation of splenocytes derived from these mice with the MHC class I (CD8^+^) restricted SIINFEKL epitope revealed the stimulation of OVA-specific cytotoxic T cell responses as indicated by increased numbers of SIINFEKL-specific T cells secreting IFNγ compared to splenocytes from mice vaccinated with OVA or chitosan-OVA nanoparticles alone (Figure 8). 

## 5. Conclusions

Chitosan nanoparticles could be prepared reproducibly by ionic gelation and incorporate 10% OVA as a model antigen. Storage stability was achieved by transfer to a dry powder Nano-in-Microparticulate (NiM) formulation, which is capable of direct administration by means of a dry powder inhaler or by administration by a microsprayer after redispersion, and this formulation was used for vaccination in the in vivo experiments. The in vivo results showed that chitosan nanoparticles co-administered with mucosal adjuvants c-di-AMP are a promising antigen delivery system for mucosal vaccination and evoke both humoral and cellular immune responses in vivo. Altogether, the vaccination strategy induced an enhanced Th1/Th17-dominated cellular immune response. Based on the observed cellular immune responses, it seems that this antigen delivery platform might be amenable for implementation in other indications for which this response profile is highly desirable (e.g., immunotherapy of lung-associated cancers and prevention of bacterial infection). For example, Van Dis et al. have demonstrated that i.n. immunization of mice using a protein subunit vaccine adjuvanted with a cyclic di-nucleotide elicited long-lasting protective immunity to *Mycobacterium tuberculosis* by stimulating a Th1/Th17 response [94]. Here, we demonstrated that c-di-AMP is a promising mucosal adjuvant for this nanoparticle-based formulation, being able to induce a protective immune response. Besides this, the enhanced dose-sparing capacity is an interesting feature for the development of innovative vaccines against respiratory pathogens. Overall, the vaccine platform presented here could be a useful tool for fighting different types of pathogens, such as viruses, bacteria and parasites. 

## Figures and Tables

**Figure 1 pharmaceutics-15-01238-f001:**
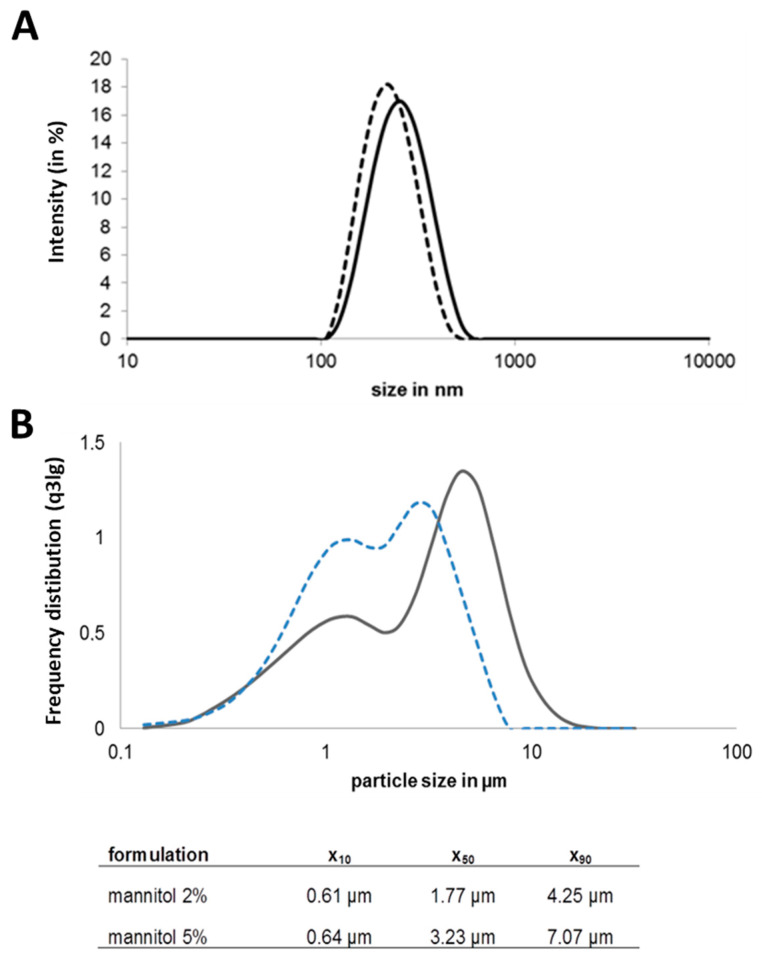
(**A**) Size distribution of OVA-loaded chitosan nanoparticles before spray drying (dotted line), after drying and redispersion in acetic acid (continuous line) (*n* = 3, mean plotted); (**B**) Particle size distribution of dried powder with different amounts of mannitol matrix; light blue line: 2% mannitol, dark grey line: 5% mannitol (one representative measurement).

**Figure 2 pharmaceutics-15-01238-f002:**
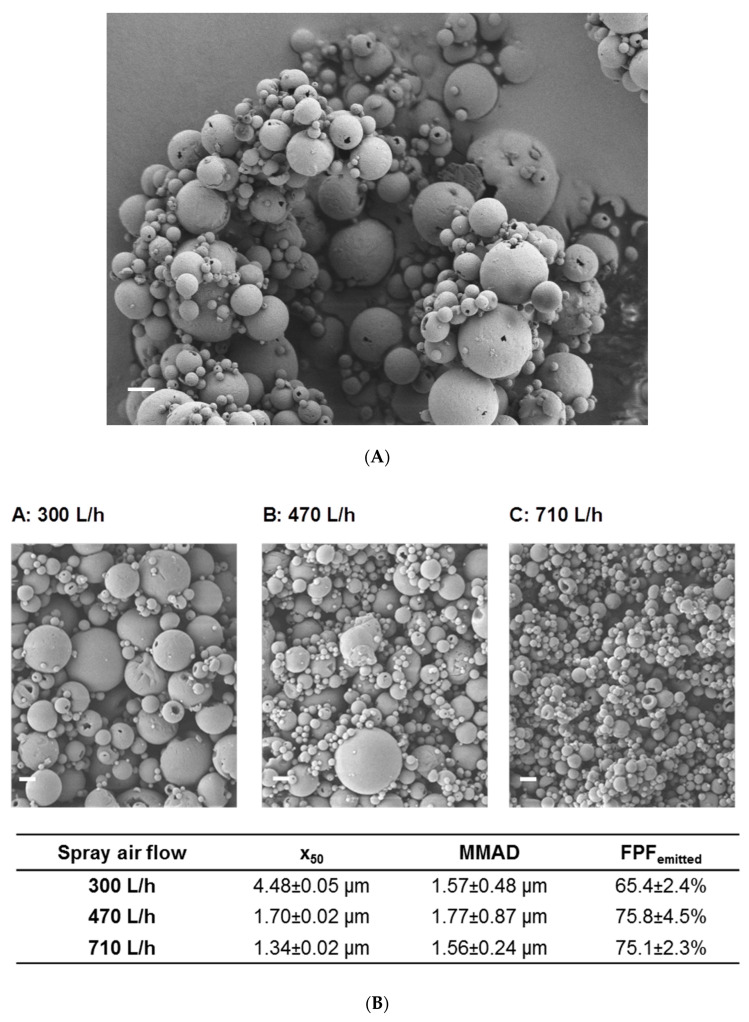
(**A**) SEM picture of optimized dry powder NiM formulation for the in vivo study (bar represents 2 µm). (**B**) Influence of the spray airflow on particle size and the corresponding fine particle fraction. Top: SEM pictures of particles spray dried with different spray air flow velocities (A–C, bar represents 2 µm), bottom: x_50_-values, FPF_emitted_ and MMAD.

**Figure 3 pharmaceutics-15-01238-f003:**
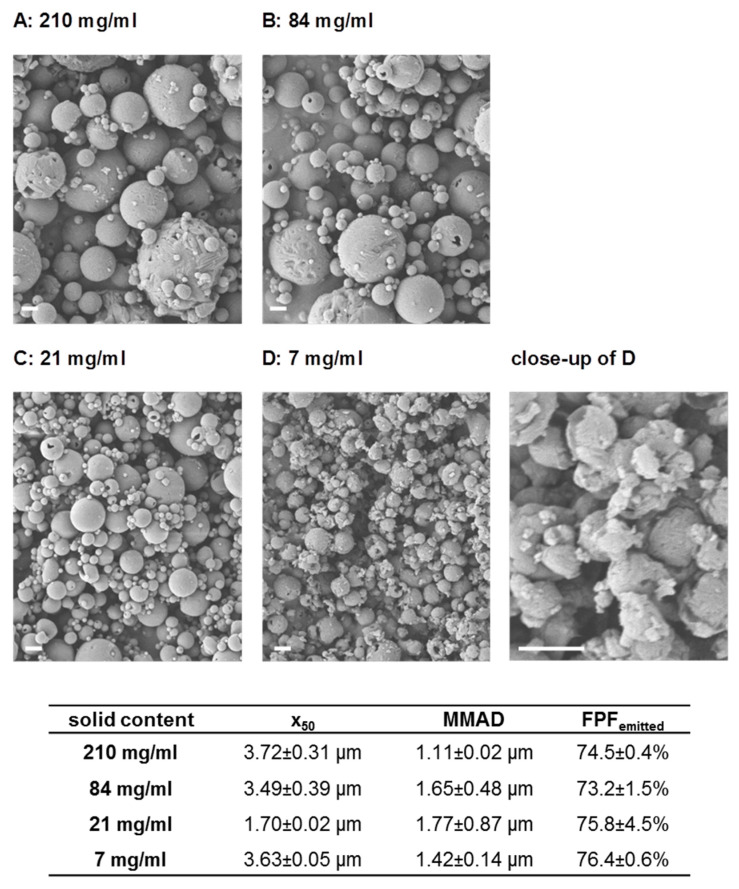
Influence of solid content on particle size and fine particle fraction. Top: SEM pictures of particles dried with various solid contents ((**A**–**D**), bar represents 2 µm); bottom: x_50_-values, FPF_emitted_ and MMAD.

**Figure 4 pharmaceutics-15-01238-f004:**
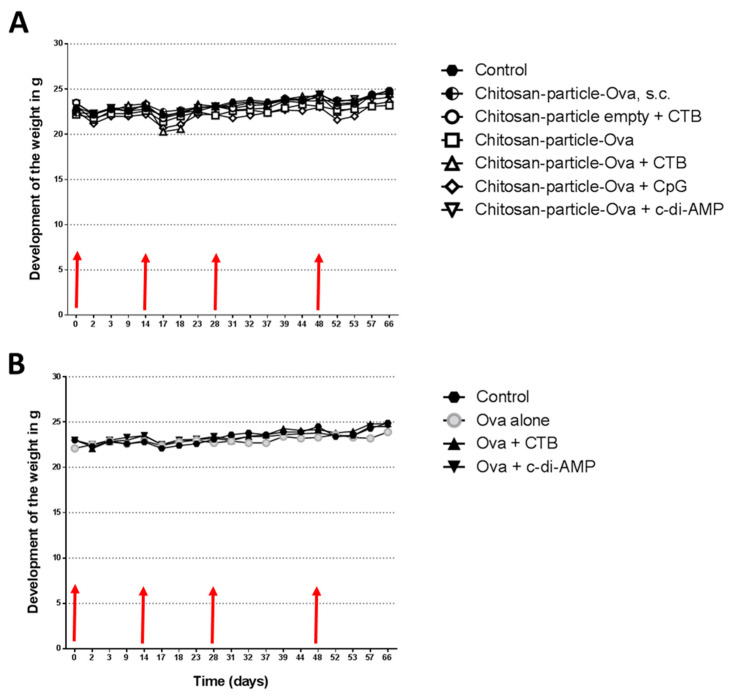
Development of the weight. No loss of weight was observed in animals receiving Ova protein and chitosan-OVA nanoparticles, respectively, by s.c. or pulmonary route. (**A**) The use of adjuvants CTB (△) or CpG (◇) in combination with chitosan-OVA nanoparticles resulted in a slight decrease in weight within 3 days after vaccination, but animals recovered within 5 days. Opposite to this, mice receiving c-di-AMP (▽) as an adjuvant showed a better general tolerance, with no or only marginal loss of body weight following vaccination comparable to values observed in untreated control animals. (**B**) Mice immunized with OVA protein co-administered with CTB (▲) and c-di-AMP (▼), respectively, showed only very mild weight loss.

**Figure 5 pharmaceutics-15-01238-f005:**
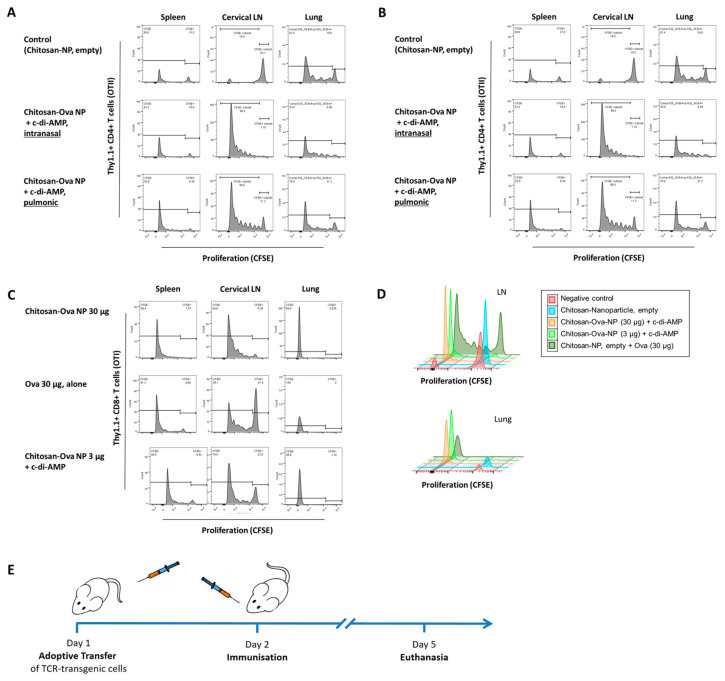
Histograms showing cell proliferation tracked by CFSE dilution in the lung, draining lymph nodes and spleen. Every cell division is indicated by the reduction of the CFSE signal strength (~50%). The shown percentages indicate the ratio of proliferated live CD4^+^ (**A**,**B**) or CD8^+^ T cells (**C**,**D**). Strategy for the adoptive transfer experiments. For the identification of the transferred TCR transgenic T cells in the wild-type C57BL/6 host, Thy1.1/OTI or OTII mice were used as donors (**E**).

**Figure 6 pharmaceutics-15-01238-f006:**
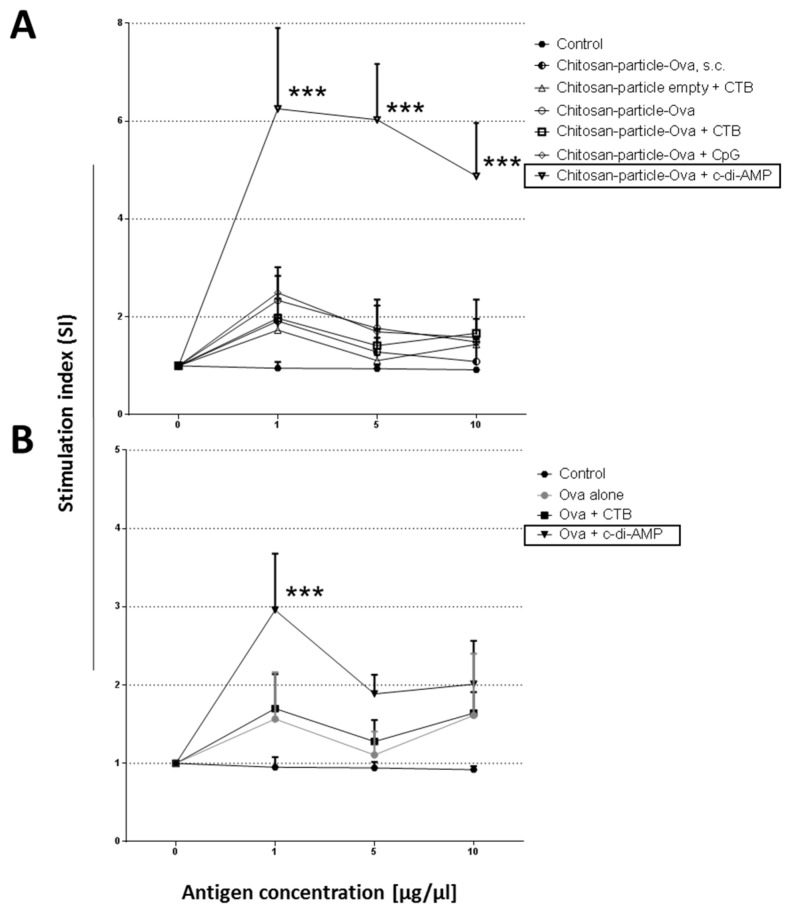
Analysis of proliferative responses in mice vaccinated with (**A**) chitosan-OVA nanoparticles (30 μg OVA/dose) or (**B**) OVA co-administered with or without 10 μg/dose of c-di-AMP (▽), CTB (△) or CpG (◇). Splenocytes from groups of mice immunized by pulmonary or s.c. route were re-stimulated for 96 h with different concentrations of OVA (1, 5 and 10 µg/mL). Untreated mice served as control, reflecting the baseline stimulation capacity of LPS-free OVA. The results are presented by stimulation index (SI) being the ratio of [^3^H]-thymidine uptake of stimulated versus non-stimulated samples. The sem is indicated by vertical lines. Differences were statistically significant at *p* < 0.001 (***) compared with splenocytes derived from mice immunized with OVA or Chitosan-OVA nanoparticles alone.

**Figure 7 pharmaceutics-15-01238-f007:**
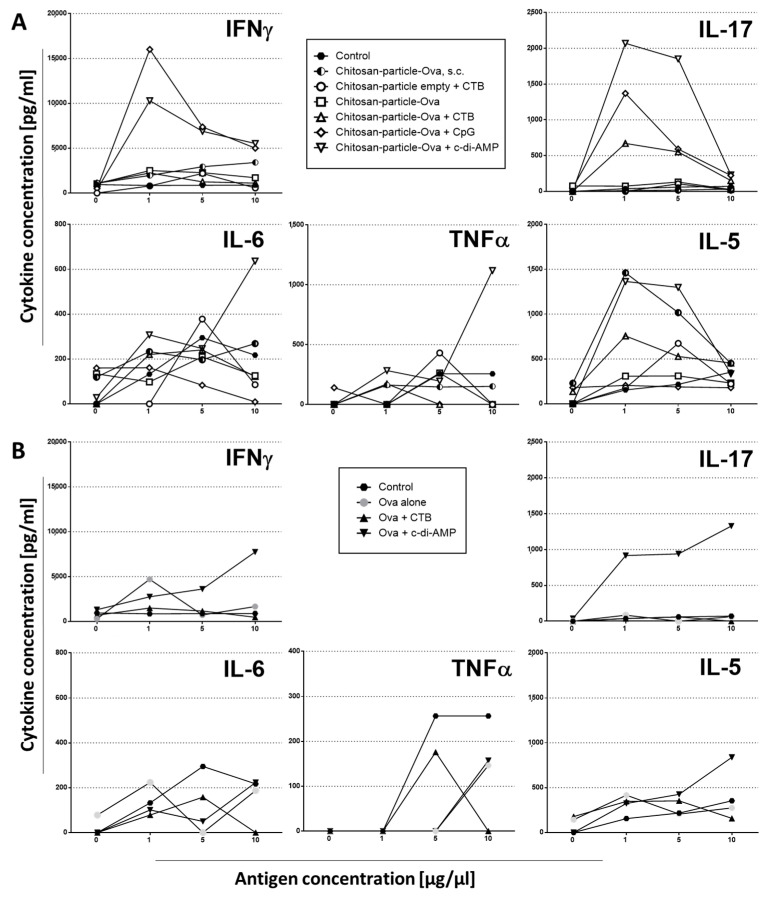
Kinetic Analysis of cytokine pattern in supernatants of antigen-restimulated splenocytes. Splenocytes of mice vaccinated by pulmonary route with (**A**) chitosan-OVA nanoparticles (30 μg OVA/dose) or (**B**) OVA co-administered with or without 10 μg/dose of c-di-AMP (▽), CTB (△) or CpG (◇). Mice immunized by s.c. route served as the “golden standard”. Splenocytes from groups of mice immunized by pulmonary or s.c. route were re-stimulated for 96 h with enhanced concentrations of OVA (1, 5 and 10 µg/mL). Untreated mice served as control, reflecting the baseline stimulation capacity of LPS-free OVA. The results are presented as cytokine concentration in pg/mL.

**Figure 8 pharmaceutics-15-01238-f008:**
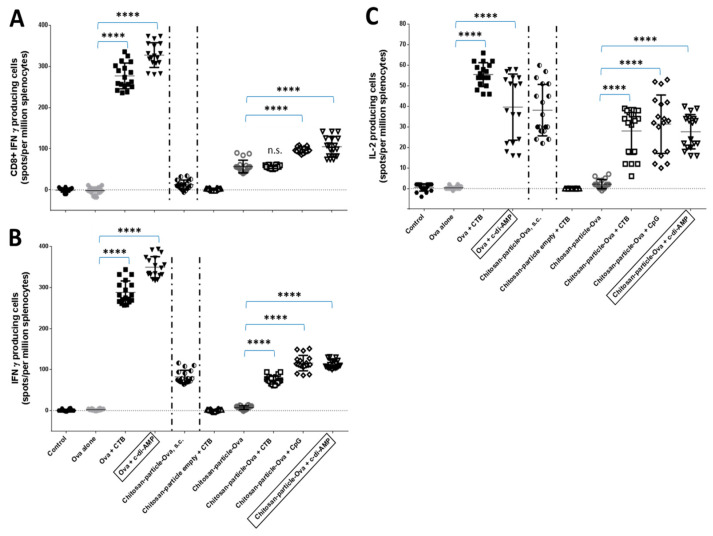
Analysis of antigen-specific cellular responses induced by vaccination. The number of IFN-γ and IL-2-producing cells was determined by ELISPOT. Splenocytes from immunized groups of mice were incubated in the presence of (**B**,**C**) OVA protein or (**A**) CD8^+^-specific OVA peptide for 24 or 48 h, respectively. Untreated mice served as control, reflecting the baseline stimulation capacity of LPS-free OVA. Results are expressed as a number of spots of cells producing cytokines per 10^6^ splenocytes. The background values of unstimulated cells were subtracted. The differences are statistically significant *p* < 0.0001 (****) compared to the results of cells obtained from mice vaccinated with OVA or chitosan-OVA nanoparticles alone.

**Figure 9 pharmaceutics-15-01238-f009:**
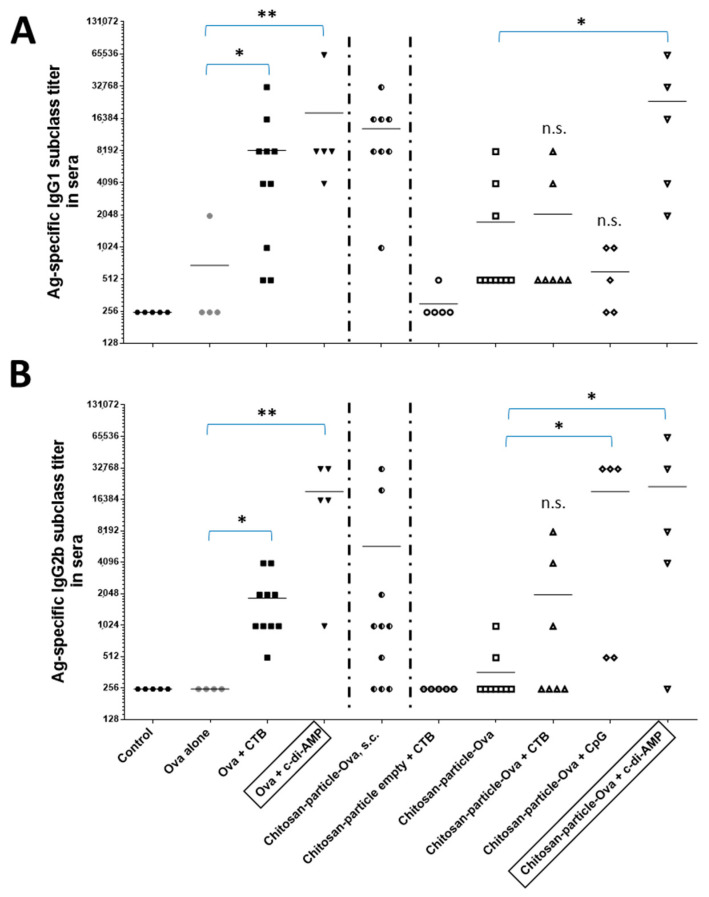
Evaluation of OVA-specific (**A**) IgG1 and (**B**) IgG2b subclass titers in sera by ELISA. C57BL/6 mice (*n* = 5 to 10) were vaccinated on days 0, 14, 28 and 48 with OVA or chitosan-OVA nanoparticles (30 μg OVA/dose) co-administered with or without 10 μg/dose of c-di-AMP (▽), CTB (△) or CpG (◇) administered in a total volume of 50 μL per dose by pulmonary route. The negative control group received only PBS. Antigen-specific IgG1 and IgG2b subclass titers were determined on day 60. The results are expressed as mean endpoint titers. Differences were statistically significant at *p* < 0.05 (*) and at *p* < 0.005 (**) with respect to values obtained in mice receiving OVA (⬤) or Chitosan-OVA nanoparticles (⬜) alone.

**Table 1 pharmaceutics-15-01238-t001:** Immunization setup for the study with antigen-naïve mice.

Group	Vaccine Formulation	Administration Route	C57BL/6	Dose per Animal
1	Control	pulmonary	5	-
2	OVA alone	pulmonary	5	30 µg/-
3	OVA + CTB	pulmonary	10	30 µg/10 µg
4	OVA + c-di-AMP	pulmonary	5	30 µg/10 µg
5	Chitosan-particle-OVA	Subcutan (s.c.)	10	30 µg
6	Chitosan-particle empty + CTB	pulmonary	5	0 µg/10 µg
7	Chitosan-particle-OVA	pulmonary	10	30 µg
8	Chitosan-particle-OVA + CTB	pulmonary	10	30 µg/10 µg
9	Chitosan-particle-OVA + CpG	pulmonary	5	30 µg/10 µg
10	Chitosan-particle-OVA + c-di-AMP	pulmonary	5	30 µg/10 µg

**Table 2 pharmaceutics-15-01238-t002:** Immunization setup for the adoptive transfer study.

Group	Vaccine Formulation	Route	C57BL/6	Number of Animals
1	Control	i.n./pulmonary	OTII	1
2	Chitosan NP, empty	pulmonary	OTII	3 *
3	OVA 30 µg	i.n./pulmonary	OTII	5
4	Chitosan-OVA NP 30 µg + c-di-AMP	i.n./pulmonary	OTII	5
5	Chitosan-OVA NP 30 µg	i.n./pulmonary	OTI	5
6	Chitosan-OVA NP 30 µg	i.n./pulmonary	OTI	3 *
7	Chitosan-OVA NP 3 µg + c-di-AMP	i.n./pulmonary	OTI	3 *

* = samples of animals of the same group were pooled prior to analysis.

## Data Availability

Data will be made available upon request.

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
