# Peer review of "Pulmonary Application of Novel Antigen-Loaded Chitosan Nano-Particles Co-Administered with the Mucosal Adjuvant C-Di-AMP Resulted in Enhanced Immune Stimulation and Dose Sparing Capacity"

_pharmaceutics, 2023, doi:10.3390/pharmaceutics15041238_

Round 1

Reviewer 1 Report

The Authors provide information of the use of chitosan nanoparticles in conjunction with cyclic-di-AMP administration as a possible vaccine platform via pulmonary route of administration. Nanoparticle formulation and in vivo data are provided along with use of adoptive transfer studies. There is interesting data presented within this work; however, the organization of the manuscript is difficult to follow and often data are presented in an order that is contrary to the text. Moreover, there is an emphasis on the in vivo data with no accompanying mechanism provide. This contrasts with the use of STING in the title with no data discerning the role of this factor in the results. A series of cell assay experiments would greatly benefit the presented in vivo data. Collectively, this work requires additional information and data to support the conclusions.

Figure 1A – Please clarify this image. Three experiments are referenced, but it is unclear if this is a representative image from the experiments.

What was the particle size generated in the absence of OVA?

Could the Authors provide information regarding the number of OVA per nanoparticle? In addition, what is the final ovalbumin concentration within the nanoparticle?

What was the impact of 2% and 5% mannitol on the zeta potential of particles?

Figure 2AB – Please confirm the images and scale bars are correct. The image for 710L/h appears quite different from the 300L/h and 470L/h images. Can you the Authors speculate on this visible difference relative to the tabulated data from the images? Likewise for Figure 3 since the scale bars appear relatively different compared to the particles within the images, but the tabulated data shows minimal differences.

Lines 274-277 – please revise this sentence since it is difficult to follow.

Figure 3 – what was the spray air flow for these experiments?

It would benefit results section 3.1 if additional information was added to connect Figures 1 through 3.

There is no description of SEM experiments. Please provide this information (e.g., instrument, sample preparation, number of images obtained, etc)

What were the final conditions used to generate the nanoparticles for subsequent in vivo studies?

Could the Authors provide rationale for why only one empty particle was performed with a single adjuvant (group 6, Table 1)?

What was the gender distribution for in vivo studies?

Could the Authors provide a zoom-in view of the y-axis for Figure 4AB? Conversely, a table summarizing percent weight change relative to control would benefit the presentation of data.

Lines 292-295 – this appears due to the use of the chitosan particles in combination with the use of CTB or CpG as opposed to either adjuvant alone.

Could the Authors comment on the comparison of the empty particle with CTB versus OVA loaded with CTB?

For Figure 4 and accompanying text, please specify the route of administration.

Please provide a diagram of the adoptive transfer experimental design.

Lines 307-310 – the order of data presented does not fit with this statement. For clarity, please organize the data presentation in a linear format with information in the text.

Lines 361-363 – could the Authors provide a quantitative assessment for this conclusion?

Lines 369-371 – that data shown in Figure 6 appears to show that proliferation was achieved (above control levels) for mice receiving OVA or chitosan-OVA NP alone or co-administered with CTB or CpG. However, there is no statistical comparison of control to all groups. Instead, the Authors only statistically compared a selected group. Please provide a complete statistical comparison among the groups.

Figure 7 – please show or mention the error bars for these data.

Figure 7 legend – please correct the letter designations for the data displayed.

Figure 8 and accompanying text – please provide a complete statistical comparison among the groups.

Could the Authors comment on the variability of the data presented in Figure 9? Please provide statistical data supporting conclusions.

Lines 604-605 – please provide relevant stability data that supports this statement.

Have the Authors tested whether other immune stimulants are equally low or absent among test articles used for cytokine studies (i.e., endotoxin contamination, etc)?

Have STING knock-out mice been considered for use? Contrary to the title, there is no connection within the data to the role of these test articles in vivo and the contribution by STING the results.

Author Response

Reviewer 1:

The Authors provide information of the use of chitosan nanoparticles in conjunction with cyclic-di-AMP administration as a possible vaccine platform via pulmonary route of administration. Nanoparticle formulation and in vivo data are provided along with use of adoptive transfer studies. There is interesting data presented within this work; however, the organization of the manuscript is difficult to follow and often data are presented in an order that is contrary to the text. Moreover, there is an emphasis on the in vivo data with no accompanying mechanism provide. This contrasts with the use of STING in the title with no data discerning the role of this factor in the results. A series of cell assay experiments would greatly benefit the presented in vivo data. Collectively, this work requires additional information and data to support the conclusions.

Figure 1A – Please clarify this image. Three experiments are referenced, but it is unclear if this is a representative image from the experiments.

For the DLS measurement the plotted line is the mean of 3 individual measurements. This has been added to the figure legend.

What was the particle size generated in the absence of OVA?

In the absence of OVA, the nanoparticles were slightly larger (50-70 nm). This can be explained by the introduction of another molecule (OVA) which is capable for ionic interactions. This increases the overall interactions and leads to a more dense and thus smaller particle.

Could the Authors provide information regarding the number of OVA per nanoparticle? In addition, what is the final ovalbumin concentration within the nanoparticle?

The exact number of molecules is unknown to the authors. The OVA concentration in the nanoparticles was 10% as stated in the results. Line 283-284

What was the impact of 2% and 5% mannitol on the zeta potential of particles?

The zeta potential of the nanoparticles was not affected by the concentration of the matrix being used for embedding in spray drying. As the matrix is not part of the particles and zeta potential is measured in dispersion, this was also not expected. Zeta potential is, however, dependent on the ions present in the dispersion medium (e.g. buffer salts) and may thus vary largely dependent on the environment.

Figure 2AB – Please confirm the images and scale bars are correct. The image for 710L/h appears quite different from the 300L/h and 470L/h images. Can you the Authors speculate on this visible difference relative to the tabulated data from the images? Likewise for Figure 3 since the scale bars appear relatively different compared to the particles within the images, but the tabulated data shows minimal differences.

The reviewer is right and visual appearance might be a bit misleading. We have carefully checked the scale and allocation of the images and it is correct. In Fig 2A the scale bar is larger than for the rest of the images as it is a different magnification. Thus, particles appear to be as large as for the 300 L/h flow, but actually are similar to the 470 L/h. In image Fig.2B  a different excerpt can be seen and it is visible (also in Fig 2A) that there is a certain particle size distribution. Thus, depending on the excerpt for the image, optical sizing may be misleading. Therefore, we rely on the laser diffraction experiments, and only used SEM for confirmation of size differences.

Lines 274-277 – please revise this sentence since it is difficult to follow.

We rephrased the corresponding paragraph.” The ratio of antigen to matrix in the final spray dried product determines the dose in mg powder to be administered and thus needs to be taken into account for administration in an in vivo small animal setup. Finally, the spray drying parameters can be used to tune particle characteristics such as size and morphology at a fixed composition of the formulation. With this it is possible to fine-tune aerodynamic performance and regional deposition if desired.” Line 279-282

Figure 3 – what was the spray air flow for these experiments?

The spray air flow for this set of experiments was 470 L/h. This has been clarified in the text. Line 320-321

It would benefit results section 3.1 if additional information was added to connect Figures 1 through 3.

A general paragraph has been introduced at the beginning of the results section to put the findings regarding the drug delivery system into perspective: “The Nano-in-Microparticle (NiM) formulation for dry powder delivery of the model antigen via inhalation has been optimised for its purpose. The antigen, ovalbumin (OVA) in this study, needs to be associated to a nanoparticulate carrier to allow specific immune targeting and cellular uptake 1. This renders nanoparticle size, antigen content and nanoparticle stability upon redispersion important characteristics. In order to obtain a storage-stable form of the nanoparticulate vaccine, it has been embedded in a dry mannitol matrix. The ratio of antigen to matrix in the final spray dried product determines the dose in mg powder to be administered and thus needs to be taken into account for administration in an in vivo small animal setup. Finally, the spray drying parameters can be used to tune particle characteristics such as size and morphology at a fixed composition of the formulation. With this it is possible to fi-ne-tune aerodynamic performance and regional deposition if desired.” Lines 272-283

There is no description of SEM experiments. Please provide this information (e.g., instrument, sample preparation, number of images obtained, etc)

Thanks for spotting this. The Scanning Electron Microscopy (SEM) method has been added to the methods section. Line 161-165.

What were the final conditions used to generate the nanoparticles for subsequent in vivo studies?

Nanoparticles were prepared as stated in the Materials and Methods section and this process has not been adapted in the course of this study. For the in vivo formulation, the nanoparticles were spray dried to obtain a storage-stable dry powder formulation. For this, 2% mannitol were added to the dispersion resulting in a spray feed content of 21 mg/ml. The dispersion was spray dried at 80 °C inlet temperature with 470 L/h and an outlet temperature of about 35 °C. As this information needs to be collected from the previous manuscript version, a paragraph summarising these parameters has been added to the results. Line 310-313.

Could the Authors provide rationale for why only one empty particle was performed with a single adjuvant (group 6, Table 1)?

The focus here was on the empty Chitosan-particle rather than on the adjuvants. We and others demonstrated already before, that the adjuvant without antigen does not increase the immunogenicity of the delivery vehicle 2,3. Furthermore, it has been shown that chitosan-particles modulate immune responses due to their intrinsic adjuvant properties 4. Moreover, including only one of such control groups allowed us to reduce the number of animals needed which is in line with the 3 R principles (refinement, replacement, reduction).

What was the gender distribution for in vivo studies?

We used only female mice. This has been added to the methods. Line 171-173

Could the Authors provide a zoom-in view of the y-axis for Figure 4AB? Conversely, a table summarizing percent weight change relative to control would benefit the presentation of data.

We adapt the text accordingly and include additional information of the weight change: “However, all the animals recovered within 5 days after each immunization and weight loss never reached more than 6 % (c-di-AMP) of the starting weight, in opposite to standard adjuvants, such as CTB (>12 %) or CpG (>8%), which showed enhanced weight loss after immunization.” Line 337-340

Lines 292-295 – this appears due to the use of the chitosan particles in combination with the use of CTB or CpG as opposed to either adjuvant alone.

We rephrased the corresponding paragraph in order to clarify this issue. Line 336-344

Could the Authors comment on the comparison of the empty particle with CTB versus OVA loaded with CTB?

We rephrased the corresponding paragraph accordingly.

For Figure 4 and accompanying text, please specify the route of administration.

The route of administration is given in line 334 and a more detailed description is given in the Material and Method section. Line 180-182 / Line 196-201

Please provide a diagram of the adoptive transfer experimental design.

We included the requested diagram in Figure 5 E in the results section. Line 380

Lines 307-310 – the order of data presented does not fit with this statement. For clarity, please organize the data presentation in a linear format with information in the text.

We revised the corresponding text accordingly. Line 310-313

Lines 361-363 – could the Authors provide a quantitative assessment for this conclusion?

We rephrased the corresponding paragraph.

Lines 369-371 – that data shown in Figure 6 appears to show that proliferation was achieved (above control levels) for mice receiving OVA or chitosan-OVA NP alone or co-administered with CTB or CpG. However, there is no statistical comparison of control to all groups. Instead, the Authors only statistically compared a selected group. Please provide a complete statistical comparison among the groups.

We rephrased the corresponding paragraph: “In contrast, no proliferation was observed in control animals (Fig. 5 A). Co-administration of chitosan-OVA nanoparticles with c-di-AMP by i.n. and pulmonary route, respectively, resulted in the induction of strong proliferation of Thy1.1/CD4+ T cells in the cervical draining LNs, shown by loss of CFSE. In opposite, samples derived from mice treated with empty chitosan-nanoparticles showed only weak unspecific proliferation (Fig. 5 B). ….” Line 389-400

Figure 7 – please show or mention the error bars for these data.

We rephrased the corresponding paragraph and the figure legend.” Figure 7. Kinetic Analysis of cytokine pattern in supernatants of antigen-restimulated splenocytes.“ Line 468-474, 483-491

Figure 7 legend – please correct the letter designations for the data displayed.

The letter designations were corrected.

Figure 8 and accompanying text – please provide a complete statistical comparison among the groups.

We rephrased the figure legend. ”The background values of unstimulated cells were subtracted. The differences are statistically significant p<0.0001 (****) compared to the results of cells obtained from mice vaccinated with OVA or chitosan-OVA nanoparticles alone.” and include statistical comparison among the adjuvanted groups. Notice: Not all statistical data is shown in figure 7, but mentioned in the corresponding paragraph. Line 499-503

Could the Authors comment on the variability of the data presented in Figure 9? Please provide statistical data supporting conclusions.

We rephrased the corresponding paragraph and the figure legend ” Antigen-specific IgG1 and IgG2b subclass titers were determined in sera samples derived from vaccinated animals included in a proof of principle animal study at day 68 (Fig. 9 A/B). There were statistically significant differences (CTB, p<0.05 (*) / c-di-AMP p<0.005 (**)) in the levels of IgG1 and IgG2b titers elicited in mice vaccinated by pulmonary route. Co-administration of CTB or CpG as positive control mucosal adjuvants resulted in the induction of higher levels of IgG2b titers (>6- and >50-fold increase, respectively) with respect to values obtained in mice receiving OVA or chitosan-OVA nanoparticles alone at p<0.05 (*). On the other hand, immunization with OVA co-administered with c-di-AMP resulted in significantly at p<0.005 (**) enhanced OVA-specific IgG1 and IgG2b subclass titers when given by pulmonary route (Fig. 9 A/B).” Line 512-520

“Also, in comparison to animals vaccinated by s.c. route, the analysis of the IgG sub-class titres obtained using c-di-AMP as adjuvant showed an increase, but less prominent with 2-fold and >3-fold increases for IgG1 and IgG2b, respectively (Fig. 9 A/B). The fact that chitosan-OVA nanoparticles alone stimulated stronger antigen-specific humoral re-sponses when given by s.c. route rather than by pulmonary route compared to OVA pro-tein alone most probably is because some part of the vaccine formulation in the lung couldn´t pass the mucosa and/or got degraded by physicochemical (e.g. pH) or enzymatic processes. In contrast, injection usually results in application of 100% of the formulation. This highlights once more the necessity of adjuvants being included in mucosal subunit vaccine formulations.” Line 526-535

We also included the statistical data into Figure 9 according to the reviewer.” Differences were statistically significant at p<0.05 (*) and at p<0.005 (**) with respect to values obtained in mice receiving OVA (˜) or Chitosan-OVA nanoparticles (£) alone.“ Line 542-544

Lines 604-605 – please provide relevant stability data that supports this statement.

The storage stability in terms of nanoparticle size has been assessed in this study (Fig. 1). The stability of the antigen in this formulation has been assessed as reported earlier 5,6. This has been referenced in the introduction (line 121-122).

Have the Authors tested whether other immune stimulants are equally low or absent among test articles used for cytokine studies (i.e., endotoxin contamination, etc)?

Both, the OVA protein as well as the MHC class I-restricted OVA peptide (SIINFEKL) used for restimulation of splenocytes derived from immunized mice were LPS-free. We included this information in the material and method section.

Have STING knock-out mice been considered for use? Contrary to the title, there is no connection within the data to the role of these test articles in vivo and the contribution by STING the results.

The direct contribution of c-di-AMP was in fact not aim of the present work. We just wanted to mention that cyclic di-nucleotides are acting via the STING pathway. In order to avoid wrong expectations we remove it from the title.

References:

1          De Temmerman, M.-L. et al. Particulate vaccines: on the quest for optimal delivery and immune response. Drug Discovery Today 16, 569-582 (2011).

2          Mittal, A. et al. Inverse micellar sugar glass (IMSG) nanoparticles for transfollicular vaccination. J Control Release 206, 140-152, doi:10.1016/j.jconrel.2015.03.017 (2015).

3          Mittal, A. et al. Efficient nanoparticle-mediated needle-free transcutaneous vaccination via hair follicles requires adjuvantation. Nanomedicine 11, 147-154, doi:10.1016/j.nano.2014.08.009 (2015).

4          Mudgal, J., Mudgal, P. P., Kinra, M. & Raval, R. Immunomodulatory role of chitosan-based nanoparticles and oligosaccharides in cyclophosphamide-treated mice. Scand J Immunol 89, e12749, doi:10.1111/sji.12749 (2019).

5          Diedrich, A. Entwicklung einer nanopartikulären Formulierung zur Vakzinierung über den Respirationstrakt Kiel University, (2015).

6          Diedrich, A. & Scherließ, R. in DDL 25    (Edinburgh, Scotland, 2014).

Reviewer 2 Report

The manuscript explored the immunogenicity of an antigen-loaded chitosan dry powder nano-in-microparticles. It is interesting work. However, the authors need to justify what is the novelty of the work in the manuscript, as mentioned in the title.

Line 131-138: Please report the drug loading and encapsulation efficiency.

Line 139-145: How’s the stability of the lipid nanoparticles and the protein after spray drying? Is the nano-particle size still retained?

Line 147-153: BP and USP suggested using a 4kpa pressure drop and 4L inhaled air for aerosol testing. Please justify if the 100l/min obeys this guideline. Also, explain how the drug was recovered from the nanoparticles.

Line 235, 246 and 351: Please revise. SEM refers to “scanning electron microscope” as well as “standard error of the mean”. Also revise “+/-“ to “±”.

Line 251-255, 264-268 and 283-286: The authors should include these experiments in the methodology because it is a little hard to follow the results section.  

Line 256-262: Most particles have holes, and they may be porous, which may lead to low density and enhanced aerosol performance. Any explanation for this?

Line 364-386: Why chitosan nanoparticles alone was not tested to justify that the particles are the one that leads to enhanced immunogenicity? Any preliminary data to justify this? 

Author Response

Reviewer 2:

The manuscript explored the immunogenicity of an antigen-loaded chitosan dry powder nano-in-microparticles. It is interesting work. However, the authors need to justify what is the novelty of the work in the manuscript, as mentioned in the title.

Line 131-138: Please report the drug loading and encapsulation efficiency.

OVA loading was 10% (w/w) as reported in Line 283-284.

This equals a loading efficiency of about 28%. Attempts to increase the loading efficiency have been undertaken in the course of AA’s project, but were not successful for the polymers used here (https://macau.uni-kiel.de/receive/diss_mods_00016827).

Line 139-145: How’s the stability of the lipid nanoparticles and the protein after spray drying? Is the nano-particle size still retained?

In this study, polymeric nanoparticles composed of chitosan and carmellose were used to encapsulate the antigen. Nanoparticle size was retained during spray drying and redispersion as shown in Figure 1A. Antigen integrity has not been checked specifically in this study 5,6 . It cannot be said that a specific structure needs to be maintained to serve as antigen. As long as the epitope gets presented, it is functional. Therefore, immunological assessment was performed. Line 147-150

Line 147-153: BP and USP suggested using a 4kpa pressure drop and 4L inhaled air for aerosol testing. Please justify if the 100l/min obeys this guideline. Also, explain how the drug was recovered from the nanoparticles.

Aerodynamic assessment was performed according to Ph.Eur. 2.9.18. This has been added in the method section. As the Cyclohaler was used, which is a low resistance device, 4 kPa pressure drop would only be achieved at flow rates above 100 L/min. The Ph.Eur. sets a maximum flow at 100 L/min. The compendial standard requests 4 L of air to be sucked through the device which results in a run time of 2.4 sec. Line 155-159

The drug was recovered from the nanoparticles by dissolving the NP and assessing protein content by Micro BCA assay. This has been added to the methods.

Line 235, 246 and 351: Please revise. SEM refers to “scanning electron microscope” as well as “standard error of the mean”. Also revise “+/-“ to “±”.

Thank you for spotting this, this has been changed accordingly.

Line 251-255, 264-268 and 283-286: The authors should include these experiments in the methodology because it is a little hard to follow the results section.

The reviewer is right, it might come as a surprise in the results section as it has not been mentioned before. A paragraph was included in the methods section: “In an additional experimental design, the spray parameters were varied in terms of spray air flow and spray feed solid content to gain understanding of tuning possibilities in case particle size needs to be optimised for targeting to specific areas in the lung.” Line 147-150

Line 256-262: Most particles have holes, and they may be porous, which may lead to low density and enhanced aerosol performance. Any explanation for this?

The particles exhibit typical morphology for spray dried materials. Due to early shell formation, particles get porous and/or hollow. This can be tailored by spray drying parameters altering the drying kinetics. This, however, was not the scope of the presented study.

Line 364-386: Why chitosan nanoparticles alone was not tested to justify that the particles are the one that leads to enhanced immunogenicity? Any preliminary data to justify this? 

There is evidence from other studies showing that pure chitosan particles may unspecifically activate the immune system, but do not cause any specific immune response 7. As mentioned in Table 1 we included Group 2, 5 and 7 which received Ova alone or Chitosan-particle-Ova via parenteral or pulmonic route to determine the base level of the induced humoral and cellular immune response. In addition, we also tested empty Chitosan-particle in an adoptive transfer model as mentioned in Table 2, Group 2. Taken together, all non-adjuvanted groups showed weaker immune responses in comparison to adjuvanted groups.

References:

7          Scherliess, R. et al. First in vivo evaluation of particulate nasal dry powder vaccine formulations containing ovalbumin in mice. Int J Pharm 479, 408-415, doi:10.1016/j.ijpharm.2015.01.015 (2015).

Reviewer 3 Report

The authors prepared a well-written and well-organized research manuscript aiming to assess whether chitosan nanocarriers in combination with the mucosal adjuvant c-di-AMP are a promising technology platform for the development of innovative mucosal vaccines against respiratory pathogens. Overall, this manuscript signifies an effort to provide a proof-of-principle study to evaluate the efficacy of chitosan nanocarriers loaded with the model antigen Ovalbumin (OVA) co-administrated with the STING agonist bis-(3’,5’)-cyclic dimeric adenosine monophosphate (c-di-AMP) given by pulmonary route.

I found that the topic of the current MS is adequate and acceptable for the journal’s scope. Even more, the manuscript addresses an important topic in the field of immunology and infectious diseases. The manuscript is well-prepared and written at a proper academic level to meet the high standards of Pharmaceutics. To this end, I recommend the manuscript be considered for publication in the journal. Nevertheless, I have some concerns/comments related to the contents written in the manuscript.

1. While the authors have argued for the importance of this topic, the novelty of this manuscript remains overlooked. The introduction section written in the manuscript is quite vague to provide an insight into why the authors decided to focus on the development of such a platform.

2. Off all mucosal adjuvants, why did the authors decide to use a STING agonist in the formulation?

3. STING agonists have been known to induce NK-medicated protective immune responses, which are also beneficial for antitumor immunity. Do the authors observe the same profile in their experimental models? In addition, STING-mediated protective immune responses can be used against bacteria as well as viruses. This means the vaccine platform can be used to fight against pathogenic bacteria in the respiratory system. Perhaps this can be inferred in the discussion section.

4. Line 51-52. The authors used “however” in two subsequent sentences. Please avoid such a thing happening by replacing one “however”.

5. Please make sure all abbreviations are written in a similar fashion. For example, decide to use one term, either elispot or ELISPOT. Some abbreviations need to be checked as well, to reflect the correct way of writing.

Author Response

Reviewer 3:

The authors prepared a well-written and well-organized research manuscript aiming to assess whether chitosan nanocarriers in combination with the mucosal adjuvant c-di-AMP are a promising technology platform for the development of innovative mucosal vaccines against respiratory pathogens. Overall, this manuscript signifies an effort to provide a proof-of-principle study to evaluate the efficacy of chitosan nanocarriers loaded with the model antigen Ovalbumin (OVA) co-administrated with the STING agonist bis-(3’,5’)-cyclic dimeric adenosine monophosphate (c-di-AMP) given by pulmonary route.

I found that the topic of the current MS is adequate and acceptable for the journal’s scope. Even more, the manuscript addresses an important topic in the field of immunology and infectious diseases. The manuscript is well-prepared and written at a proper academic level to meet the high standards of Pharmaceutics. To this end, I recommend the manuscript be considered for publication in the journal. Nevertheless, I have some concerns/comments related to the contents written in the manuscript.

  1. While the authors have argued for the importance of this topic, the novelty of this manuscript remains overlooked. The introduction section written in the manuscript is quite vague to provide an insight into why the authors decided to focus on the development of such a platform.

  1. Off all mucosal adjuvants, why did the authors decide to use a STING agonist in the formulation?

c-di-AMP has been chosen as we and others have demonstrated previously that incorporation of this adjuvant in different vaccine formulations efficiently increases antigen-specific immune responses at mucosal and systemic level. Thus, when c-di-AMP was co-administered with nanoparticulated antigens such as OVA or influenza hemagglutinin (HA) for example by intranasal route protective immune responses have been stimulated 17-19. The same is true when recombinant proteins have been combined with c-di-AMP 20-22. Finally, we have known that c-di-AMP constitutes a very robust molecule, which can by lyophilized and incorporated into a dry powder formulation without losing functionality.

We included this explanation in the discussion section. Line 634-642

  1. STING agonists have been known to induce NK-medicated protective immune responses, which are also beneficial for antitumor immunity. Do the authors observe the same profile in their experimental models? In addition, STING-mediated protective immune responses can be used against bacteria as well as viruses. This means the vaccine platform can be used to fight against pathogenic bacteria in the respiratory system. Perhaps this can be inferred in the discussion section.

We included a corresponding statement in the conclusions.

“For example, Van Dis et al. have demonstrated that i.n. immunization of mice using a protein subunit vaccine adjuvanted with a cyclic di-nucleotide elicited long-lasting protective immunity to Mycobacterium tuberculosis by stimulating a Th1/Th17 response 23. Here, we demonstrated that c-di-AMP is a promising mucosal adjuvant for this nanoparticle-based formulation, being able to induce protective immune response. Beside this, the enhanced dose sparing capacity is an interesting feature for the development of innovative vaccines against respiratory pathogens. Overall, the vaccine platform presented here could be an useful tool fighting different types of pathogens such as viruses, bacteria and parasites.“ Line 727-739

  1. Line 51-52. The authors used “however” in two subsequent sentences. Please avoid such a thing happening by replacing one “however”.

We corrected the sentence accordingly

  1. Please make sure all abbreviations are written in a similar fashion. For example, decide to use one term, either elispot or ELISPOT. Some abbreviations need to be checked as well, to reflect the correct way of writing.

We revised the abbreviations used in the present work and corrected them were appropriate.

References:

1          De Temmerman, M.-L. et al. Particulate vaccines: on the quest for optimal delivery and immune response. Drug Discovery Today 16, 569-582 (2011).

2          Mittal, A. et al. Inverse micellar sugar glass (IMSG) nanoparticles for transfollicular vaccination. J Control Release 206, 140-152, doi:10.1016/j.jconrel.2015.03.017 (2015).

3          Mittal, A. et al. Efficient nanoparticle-mediated needle-free transcutaneous vaccination via hair follicles requires adjuvantation. Nanomedicine 11, 147-154, doi:10.1016/j.nano.2014.08.009 (2015).

4          Mudgal, J., Mudgal, P. P., Kinra, M. & Raval, R. Immunomodulatory role of chitosan-based nanoparticles and oligosaccharides in cyclophosphamide-treated mice. Scand J Immunol 89, e12749, doi:10.1111/sji.12749 (2019).

5          Diedrich, A. Entwicklung einer nanopartikulären Formulierung zur Vakzinierung über den Respirationstrakt Kiel University, (2015).

6          Diedrich, A. & Scherließ, R. in DDL 25    (Edinburgh, Scotland, 2014).

7          Scherliess, R. et al. First in vivo evaluation of particulate nasal dry powder vaccine formulations containing ovalbumin in mice. Int J Pharm 479, 408-415, doi:10.1016/j.ijpharm.2015.01.015 (2015).

8          Babiuch, K., Gottschaldt, M., Werz, O. & Schubert, U. S. Particulate transepithelial drug carriers: barriers and functional polymers. RSC Adv. DOI: 10.1039/c2ra20726e, 1-39 (2012).

9          Csaba, N., Garcia-Fuentes, M. & Alonso, M. J. Nanoparticles for nasal vaccination. Adv Drug Deliv Rev 61, 140-157 (2009).

10        Vyas, S. P. & Gupta, P. N. Implication of nanoparticles/microparticles in mucosal vaccine delivery. Expert Review on Vaccines 6, 401-418 (2007).

11        Ohmes, J. et al. Injectable Thermosensitive Chitosan-Collagen Hydrogel as A Delivery System for Marine Polysaccharide Fucoidan. Mar Drugs 20, doi:10.3390/md20060402 (2022).

12        Scherliess, R. et al. Induction of protective immunity against H1N1 influenza A(H1N1)pdm09 with spray-dried and electron-beam sterilised vaccines in non-human primates. Vaccine 32, 2231-2240, doi:10.1016/j.vaccine.2014.01.077 (2014).

13        Scherliess, R. et al. In vivo evaluation of chitosan as an adjuvant in subcutaneous vaccine formulations. Vaccine 31, 4812-4819, doi:10.1016/j.vaccine.2013.07.081 (2013).

14        Walter, F. et al. Chitosan nanoparticles as antigen vehicles to induce effective tumor specific T cell responses. PloS one 15, e0239369, doi:10.1371/journal.pone.0239369 (2020).

15        Hanefeld, A. et al. Antigen-loaded chitosan nanoparticles for immunotherapy. WO2015/185180A1 (2014).

16        Wang, J. J. et al. Recent advances of chitosan nanoparticles as drug carriers. Int J Nanomedicine 6 (2011).

17        Ebensen, T. et al. Mucosal Administration of Cycle-Di-Nucleotide-Adjuvanted Virosomes Efficiently Induces Protection against Influenza H5N1 in Mice. Front Immunol 8, 1223, doi:10.3389/fimmu.2017.01223 (2017).

18        Schulze, K., Ebensen, T., Babiuk, L. A., Gerdts, V. & Guzman, C. A. Intranasal vaccination with an adjuvanted polyphosphazenes nanoparticle-based vaccine formulation stimulates protective immune responses in mice. Nanomedicine 13, 2169-2178, doi:10.1016/j.nano.2017.05.012 (2017).

19        Schulze, K. et al. Bivalent mucosal peptide vaccines administered using the LCP carrier system stimulate protective immune responses against Streptococcus pyogenes infection. Nanomedicine 13, 2463-2474, doi:10.1016/j.nano.2017.08.015 (2017).

20        Ebensen, T. et al. Bis-(3',5')-cyclic dimeric adenosine monophosphate: strong Th1/Th2/Th17 promoting mucosal adjuvant. Vaccine 29, 5210-5220, doi:10.1016/j.vaccine.2011.05.026 (2011).

21        Sanchez Alberti, A. et al. Engineered trivalent immunogen adjuvanted with a STING agonist confers protection against Trypanosoma cruzi infection. NPJ Vaccines 2, 9, doi:10.1038/s41541-017-0010-z (2017).

22        Sanchez, M. V. et al. Intranasal delivery of influenza rNP adjuvanted with c-di-AMP induces strong humoral and cellular immune responses and provides protection against virus challenge. PloS one 9, e104824, doi:10.1371/journal.pone.0104824 (2014).

23        Van Dis, E. et al. STING-Activating Adjuvants Elicit a Th17 Immune Response and Protect against Mycobacterium tuberculosis Infection. Cell Rep 23, 1435-1447, doi:10.1016/j.celrep.2018.04.003 (2018).

Reviewer 4 Report

Ebensen et al present a novel nanoparticle-mucosal adjuvant (c-di-AMP) formulation that appears to be suitable for inhalatory, dry-powder vaccination. The topic is very relevant and timely. Although a model antigen such as ovalbumin (OVA) was employed in an admittedly proof-of-principle investigation the study appears to be sufficiently thorough and the main conclusions appear reasonably well supported. Nevertheless, as specified below, I think there is room for further improvements and clarifications, mainly aimed at improving ms. readability.

Major points to be addressed:

1. Since induction of an antigen-specific humoral immune response is the minimum requirement for any vaccine, I think that moving the related section (#3.5) from the very end to the beginning of the ‘Results’ would make the overall presentation clearer and would motivate the reader to reach the end of the paper, with all the details on vaccine-induced T cell proliferation and organ-specific responses showing-up in subsequent sections.

2. The authors only mention an initial weight loss three days after priming, but, in fact, there seems to be a subsequent weight loss also around day 17, i.e., after the first boost.

3. Figure 5: why is the Chitosan-OVA NP 30 µg + c-di-AMP treatment shown in the summary/multi-axis graph in panel D, but not in panel C?

-Why not showing OVA alone (instead of Chitosan alone) in 5B?

-Why not showing a summarizing multi-axis graph also for CD4+ T-cells as well?

-The amount of OVA antigen is indicated in 5C and 5D, but not in 5A and 5B, please add.

-Is the ‘control’ in panel 5A the same as in panel 5B? If so, it should be specified.

-Were the ‘controls’ delivered intranasally or to the lungs (‘pulmonic’)?

-Were all the data in 5C (and 5D) obtained by pulmonary delivery? If so, it should be specified.

-Axes descriptors in this figure are way too small and very hard, if not impossible, to read.

-The CFSE descriptor should be added for consistency also at the bottom of panel 5A.

4. How was intranasal delivery performed? It is not mentioned in the ‘Methods’ sections, but it should be specified somewhere in the text.

5. Figure 6B (line 336) and Figure 6A (line 340) should, perhaps, be, instead, Figure 5A and Figure 5B?

If not, the authors should specify in the text that they are moving from an ‘adoptive transfer’ to a ‘radioactively labeled thymidine’ approach for measuring cell proliferation and, may be, provide some additional comment/description of the results obtained with the latter experimental set-up.

6. Also at line 340: how do the authors explain the “weak induction of cell proliferation” observed with lung and spleen samples? With what kind of treatment? Is this a ‘data not shown’? If so, it should be specified.

7. The nature of the ‘control’ utilized for the experiments in Fig. 6 should be specified.

The route of administration utilized for vaccination must also be clearly stated: in the legend (line 348) it is mentioned that subcutaneous or pulmonary delivery was employed, whereas only the pulmonary route is mentioned later on in the text (line 367).

8. At line 360, the interaction between the protein antigen and chitosan is described as ‘tight’: is there any experimental evidence for that?

9. Figure 7: in the legend, panel B is described before panel A. Why?

-The nature of the ‘control’ (PBS, Chitosan-NP empty, or something else?) is not specified and except for one treatment in 5A, for which the subcutaneous route of administration is indicated, there is no explicit mention of the mode of delivery (presumably pulmonary) for all the other treatments.

-There is no indication of intra-experiment variability in this dataset: were the experiments conducted at least in duplicate? If so, what kind of variability was observed?

-The fact that the levels of two of the cytokines shown in 7B (IL-6 and TNF) are actually higher in the ‘control’ than in the vaccinated animals should be mentioned somewhere and, if possible, at least tentatively explained.

10. The statement that the requirement for a higher amount (not concentration, as written) of OVA in order to stimulate comparable levels of IL-17 and IFN-g “demonstrates the higher potency of the nanoparticle-based formulation” (lines 381-384) must be better explained and justified.

11. To what particular type of particles do the authors refer to in the sentence at lines 384-386? All of them regardless of the adjuvant, or a subset of NPs supplemented with specific adjuvants?

12. Figure 8: the fact that soluble OVA adjuvanted with either CTB or c-di-AMP appears to produce considerably higher ELISPOT signals than the Chitosan-NP formulation must be mentioned somewhere in the text and possibly explained, at least tentatively.

-As in many other figures, the nature of the ‘control’ is not mentioned; please specify.

13. Why was the predominant effect of c-di-AMP-adjuvanted OVA chitosan-NPs on IFN-g and IL-2 expected? Should that expectation be based on the results of prior work, that work should be quoted.

14. The sentence at lines 401-402 is quite obscure and must be clarified or reworded.

15. Figure 9: why was the addition of the CpG adjuvant not included among the soluble OVA formulations (left-side data in both panels), but only in the NP datasets?

-How do the authors explain the superior performance of NP-OVA compared to soluble OVA when given subcutaneously, but not through pulmonary delivery? This should be mentioned and, if possible, at least tentatively explained somewhere in the text.

16. OVA-specific IgA responses are described as weak in NL and VL at lines 449-450, but there is no mention of IgA responses in BAL. Where they also measured? If so, where they also weak?

Minor points:

-The sentence at lines 595-597 does not read well and must be reformulated.

-A few typos throughout the ms. should be fixed.

-A list of abbreviations would be useful.

Author Response

Reviewer 4:

Ebensen et al present a novel nanoparticle-mucosal adjuvant (c-di-AMP) formulation that appears to be suitable for inhalatory, dry-powder vaccination. The topic is very relevant and timely. Although a model antigen such as ovalbumin (OVA) was employed in an admittedly proof-of-principle investigation the study appears to be sufficiently thorough and the main conclusions appear reasonably well supported. Nevertheless, as specified below, I think there is room for further improvements and clarifications, mainly aimed at improving ms. readability.

Major points to be addressed:

  1. Since induction of an antigen-specific humoral immune response is the minimum requirement for any vaccine, I think that moving the related section (#3.5) from the very end to the beginning of the ‘Results’ would make the overall presentation clearer and would motivate the reader to reach the end of the paper, with all the details on vaccine-induced T cell proliferation and organ-specific responses showing-up in subsequent sections.

It would be generally possible, but our general structure of the article attempt to show the development of a vaccine from bench to the first tests in an animal model:

  1. Development of antigen delivery system Chitosan-particle-Ova (Fig. 1 to 3)
  2. Identification of possible side effects of novel pulmonic vaccines in mice by scoring and weight (Fig. 4)
  3. Optimization of vaccines (application route, dose sparing, etc.) and their action on antigen-specific CD4+ and CD8+ T cells by using the adoptive transfer animal model (Fig. 5E)
  4. Antigen-specific cellular immune response (Fig. 6)
  5. Kinetic analysis of cytokine pattern of slpenocytes stimulated by enhanced concentration of Ova (Fig. 7)
  6. Cytokine secretion of antigen-specific CD4+ and CD8+ T cells (Fig. 8)
  7. Antigen-specific humoral immune response (Fig. 9)

  1. The authors only mention an initial weight loss three days after priming, but, in fact, there seems to be a subsequent weight loss also around day 17, i.e., after the first boost.

We apologize for the misunderstanding.

In fact, we meant within 3 days after each application. We rephrased the corresponding sentence accordingly. “The use of different adjuvants, such as CTB or CpG, resulted in a decrease in weight within 3 days after each immunization. The decrease seems to be even stronger combining chitosan nanoparticles with either adjuvant. However, all the animals recovered within 5 days after each immunization and weight loss never reached more than 6 % (c-di-AMP) of the starting weight, in opposite to standard adjuvants, such as CTB (>12 %) or CpG (>8%), which showed enhanced weight loss after immunization. Interestingly, mice immunized with empty chitosan particles + c-di-AMP also showed only minor weight loss underlining the fact that only the combination of antigen nanoparticle and adjuvant resulted in increased reactogenicity.” Line 335-347

  1. Figure 5: why is the Chitosan-OVA NP 30 µg + c-di-AMP treatment shown in the summary/multi-axis graph in panel D, but not in panel C?

Fig. 5 C/D adoptive transfer attempt to show, that

  1. pulmonic application of the Ova-protein (30 µg) or chitosan-Ova-nanoparticle (30 µg) alone or chitosan-Ova-nanoparticle (3 µg) co-administered with CDA showed the same proliferation of antigen-specific CD8+ T cells in different compartments, such as spleen, draining LN´s or lung (Fig 5 C). Thus the use of CDA showed a strong dose sparing capacity (dose reduction from 30 to 3 µg antigen).
  2. In 5 D, we included all controls (no proliferation, CFSE+) and compared the pulmonic application of chitosan-Ova-nanoparticle (30 µg) or chitosan-Ova-nanoparticle (3 µg) co-administered with CDA. We observed no differences in the induction of proliferation of antigen-specific CD8+ T cells (CFSE-) in the processed compartments (spleen, Cervical LN and lung). Whereas the application of chitosan-Ova-nanoparticle (30 µg) alone showed a weaker loss of CFSE.

-Why not showing OVA alone (instead of Chitosan alone) in 5B?

Fig. 5 A/B adoptive transfer attempt to show, that

  1. pulmonic application of the Ova-protein co-administered with CDA is comparable with intranasal application (Fig 5 A). Control samples showed only base level proliferation of antigen-specific CD4+ T cells in different compartments, such as spleen, draining LN´s or lung.
  2. Pulmonic vs. intranasal application of the chitosan-Ova-nanoparticle co-administered with CDA showed comparable results (Fig 5 B). Whereas chitosan-nanoparticle empty showed no unspecific influence on proliferation of antigen-specific CD4+ T cells in different compartments.

We included two different control samples to clearly show, that chitosan-nanoparticles will not influence the immune response in an unspecific manner. In addition, both mucosal application routes (pulmonic vs. intranasal) will train and further support the development of mucosal chitosan-nanoparticle antigen delivery system.     

-Why not showing a summarizing multi-axis graph also for CD4+ T-cells as well?

We used different graph to point to the high potential of chitosan-nanoparticles as drug delivery system for CD4+ and/or CD8+ T cells. Moreover, chitosan-nanoparticles can also act in combination with an adjuvant to induce efficient immune responses even with 10 x times lower antigen concentration. Thus, dose sparing capacity will be of high interest during pandemic.  

-The amount of OVA antigen is indicated in 5C and 5D, but not in 5A and 5B, please add.

We included the missing information in the corresponding paragraph.

“The application of 30 µg OVA co-administered with 10 µg c-di-AMP induced strong proliferation of Thy1.1+/CD4+ T cells.“ Line 386-387

“Co-administration of 3 or 30 µg chitosan-OVA nanoparticles with 10 µg c-di-AMP by i.n. and pulmonary route, respectively, resulted in the induction of strong proliferation of Thy1.1/CD4+ T cells in the cervical draining LNs shown by the loss of CFSE (Fig. 5 A, B).”  Line 390-393

-Is the ‘control’ in panel 5A the same as in panel 5B? If so, it should be specified.

No. We used different negative controls.

Fig.5 A: Mice were immunized with PBS buffer alone by the intranasal route

Fig.5 B: Chitosan-nanoparticle empty showed no unspecific influence on proliferation of antigen-specific CD4+ T cells in different compartments, such as spleen, draining LN´s or lung.

-Were the ‘controls’ delivered intranasally or to the lungs (‘pulmonic’)?

Control mice were immunized with PBS buffer alone or Chitosan-nanoparticle empty by the intranasal route.

-Were all the data in 5C (and 5D) obtained by pulmonary delivery? If so, it should be specified.

We included the missing information in the corresponding paragraph. “..at the site of administration (lung), as well as systemically (spleen) following i.n. or pulmonary administration as shown in Fig. 5 C-D.“ Line 402-406

-Axes descriptors in this figure are way too small and very hard, if not impossible, to read.

We adapted the descriptions of the axes in Fig 5 according to reviewer comments.

-The CFSE descriptor should be added for consistency also at the bottom of panel 5A.

We adapted the descriptions of the axes in Fig 5 according to reviewer comments.

  1. How was intranasal delivery performed? It is not mentioned in the ‘Methods’ sections, but it should be specified somewhere in the text

We included the missing information in the Material & Method section 2.5. “The tip of the device was gently inserted down the trachea of the anesthetized animal and 50 µl of air-free plume of liquid aerosol was administered directly into the lungs.” Line 180-182

  1. Figure 6B (line 336) and Figure 6A (line 340) should, perhaps, be, instead, Figure 5A and Figure 5B?

Thank you for the notification. We changed the text accordingly.

If not, the authors should specify in the text that they are moving from an ‘adoptive transfer’ to a ‘radioactively labeled thymidine’ approach for measuring cell proliferation and, may be, provide some additional comment/description of the results obtained with the latter experimental set-up.

We included the missing information in the corresponding paragraph.

  1. Also at line 340: how do the authors explain the “weak induction of cell proliferation” observed with lung and spleen samples? With what kind of treatment? Is this a ‘data not shown’? If so, it should be specified.

We rephrased the corresponding paragraph in order to increase comprehensibility.

  1. The nature of the ‘control’ utilized for the experiments in Fig. 6 should be specified.

We included the missing information in the corresponding figure legend. Line 439-440

The route of administration utilized for vaccination must also be clearly stated: in the legend (line 348) it is mentioned that subcutaneous or pulmonary delivery was employed, whereas only the pulmonary route is mentioned later on in the text (line 367).

We included the missing information in the corresponding paragraph. Line 349-356

  1. At line 360, the interaction between the protein antigen and chitosan is described as ‘tight’: is there any experimental evidence for that?

Recent studies have focused on the interaction of chitosan with proteins and ITC measurements confirmed intense interactions (data not published yet, manuscript in preparation).

  1. Figure 7: in the legend, panel B is described before panel A. Why?

The letter designations were corrected.

-The nature of the ‘control’ (PBS, Chitosan-NP empty, or something else?) is not specified and except for one treatment in 5A, for which the subcutaneous route of administration is indicated, there is no explicit mention of the mode of delivery (presumably pulmonary) for all the other treatments.

We included the missing information in the figure legend. Line 468-474

-There is no indication of intra-experiment variability in this dataset: were the experiments conducted at least in duplicate? If so, what kind of variability was observed?

We analysed chitosan-nanoparticle in 4 independent adoptive transfer models (n=4) and two independent in vivo experiments with Ova Protein versus Chitosan-nanoparticle-Ova.

-The fact that the levels of two of the cytokines shown in 7B (IL-6 and TNF) are actually higher in the ‘control’ than in the vaccinated animals should be mentioned somewhere and, if possible, at least tentatively explained.

We included the missing information in the corresponding section of the manuscript. Line 450-457

  1. The statement that the requirement for a higher amount (not concentration, as written) of OVA in order to stimulate comparable levels of IL-17 and IFN-g “demonstrates the higher potency of the nanoparticle-based formulation” (lines 381-384) must be better explained and justified.

We rephrased the corresponding paragraph in order to increase comprehensibility. Line 458- 467

  1. To what particular type of particles do the authors refer to in the sentence at lines 384-386? All of them regardless of the adjuvant, or a subset of NPs supplemented with specific adjuvants?

We rephrased the corresponding paragraph in order to increase comprehensibility. „ The analysis of the lymphoproliferative responses of splenocytes showed that chi-tosan-OVA nanoparticles co-administered with c-di-AMP () by pulmonary route were able to induce a strong lymphoproliferation after re-stimulation with 1 µg of LPS-free OVA with a statistical significant (p<0.001, ***) stimulation index (SI) of >6, (Fig. 6 A). In con-trast, no OVA-specific proliferation (SI <3), even with higher dosages (5 to 10 µg), were ob-served in mice receiving chitosan-OVA nanoparticles alone or co-administered with CTB () or CpG (). The same is true for mice immunized with empty chitosan-nanoparticles co-administered with CTB by s.c. route (SI <3). Moreover, no statistically significant prolif-eration has been observed in these mice (Fig. 6 A). The analysis of the lymphoproliferative responses of splenocytes showed that OVA protine co-administered with c-di-AMP () by pulmonary route were also able to induce a strong lymphoproliferation after re-stimulation with 1 µg of LPS-free OVA with a statisti-cal significant (p<0.001, ***) SI >3 (Fig. 6 B), whereas OVA co-administered with or without 10 μg/dose CTB () showed no statistically significant proliferation.” Line 419-432

  1. Figure 8: the fact that soluble OVA adjuvanted with either CTB or c-di-AMP appears to produce considerably higher ELISPOT signals than the Chitosan-NP formulation must be mentioned somewhere in the text and possibly explained, at least tentatively.

-As in many other figures, the nature of the ‘control’ is not mentioned; please specify.

We included the missing information in the figure legend. “Untreated mice serve as control reflecting the base line stimulation capacity of LPS-free OVA. Results are expressed as number of spots of cells producing cytokines per 106 splenocytes. The background values of unstimulated cells were subtracted. The differences are statistically significant p<0.0001 (****) compared to the results of cells obtained from mice vaccinated with OVA or chitosan-OVA nanoparticles alone.“ Line 500-504

  1. Why was the predominant effect of c-di-AMP-adjuvanted OVA chitosan-NPs on IFN-g and IL-2 expected? Should that expectation be based on the results of prior work, that work should be quoted.

Our phrasing was misleading. We adapted the corresponding paragraph accordingly. Line 483-491

  1. The sentence at lines 401-402 is quite obscure and must be clarified or reworded.

We rephrased the corresponding paragraph.

  1. Figure 9: why was the addition of the CpG adjuvant not included among the soluble OVA formulations (left-side data in both panels), but only in the NP datasets?

The Ova + CpG adjuvant was not included, due to the fact, that this experimental setting should only test in general the pulmonic application route using the Ova-antigen in combination with a standard mucosal adjuvant (CTB) and the novel c-di-AMP. For the comparison of the novel application strategy using chitosan-Ova-nanoparticle, the drug delivery system was combined with all adjuvants (CTB, CpG and c-di-AMP).  

-How do the authors explain the superior performance of NP-OVA compared to soluble OVA when given subcutaneously, but not through pulmonary delivery? This should be mentioned and, if possible, at least tentatively explained somewhere in the text.

The reason for that most probably is because some part of the vaccine formulation in the lung couldn´t pass the mucosa and/or got degraded by physicochemical (e.g. pH) or enzymatic processes. In contrast, injection usually results in application of 100% of the formulation. Highlighting once more the necessity of including an adjuvant.

We included the requested information in the corresponding section. Line 526-535

  1. OVA-specific IgA responses are described as weak in NL and VL at lines 449-450, but there is no mention of IgA responses in BAL. Where they also measured? If so, where they also weak?

Yes. The overall results for BAL, NL and VL were not encouraging. Taken together, we observed the induction of antigen-specific IgA titer in lavages samples derived from vagina flushings (500 µl PBS) in mice vaccinated with Ova and chitosan-Ova-nanoparticle co-administered  with c-di-AMP, followed by nasal and broncho alveolar lavages as shown in the figure below. Due to the high values of the SEM in VL-samples, we were not able to calculate significant differences. Thus, we just described weak local IgA titer in VL and NL.      

Minor points:

-The sentence at lines 595-597 does not read well and must be reformulated.

We reformulated the sentence in the revised manuscript.

-A few typos throughout the ms. should be fixed.

We lectured the manuscript once more in order to fix the typos.

-A list of abbreviations would be useful.

A list of abbreviations is now included according to comments of reviewer 4. Line 757-783

APC                antigen-presenting cells

c-di-AMP        bis-(3’,5’)-cyclic dimeric adenosine monophosphate

CD                  cluster of differentiation

CpG                CpG oligodeoxynucleotide.

CTB                cholera toxin B subunit

CTL                 cytotoxic T lymphocytes

DPI                 dry powder inhaler

ELISPOT        enzyme-linked immunospot

FPF                 fine particle fraction

rHA                 hemagglutinin

IgG                  immune globulin

IL-2                 interleukin 2

i.m.                  intramuscular

IFNγ                interferon gamma

i.n.                   intranasal

LN                   lymph nodes

MMAD            mass median aerodynamic diameter

MHC               major histocompatibility complex

NiM                 Nano-in-Microparticulate

NP                  nanoparticles

OVA                Ovalbumin

s.c.                  subcutaneous

Th1                 T helper cell 1

TNFα              tumor necrosis factor alpha

TLR                 Toll-like receptor

WHO              world health organisation

Round 2

Reviewer 2 Report

Thank you to the authors for their response. I recommend accepting the manuscript.

Reviewer 3 Report

The authors have addressed all my suggestions/comments. I think the manuscript has been adequately improved to be accepted in its present form. 

Reviewer 4 Report

Revision work and specific answers provided by the authors appear adequate.